# Thymic iNKT single cell analyses unmask the common developmental program of mouse innate T cells

S. Harsha Krovi[1,3], Jingjing Zhang[1,4], Mary Jessamine Michaels-Foster[1], Tonya Brunetti[1], Liyen Loh 🄭 [1], James Scott-Browne[1,2] & Laurent Gapin 🄭 [1,2✉]

Most T lymphocytes leave the thymus as naïve cells with limited functionality. However, unique populations of innate-like T cells differentiate into functionally distinct effector subsets during their development in the thymus. Here, we profiled >10,000 differentiating thymic invariant natural killer T (iNKT) cells using single-cell RNA sequencing to produce a comprehensive transcriptional landscape that highlights their maturation, function, and fate decisions at homeostasis. Our results reveal transcriptional profiles that are broadly shared between iNKT and mucosal-associated invariant T (MAIT) cells, illustrating a common core developmental program. We further unmask a mutual requirement for *Hivep3*, a zinc finger transcription factor and adapter protein. *Hivep3* is expressed in early precursors and regulates the post-selection proliferative burst, differentiation and functions of iNKT cells. Altogether, our results highlight the common requirements for the development of innate-like T cells with a focus on how *Hivep3* impacts the maturation of these lymphocytes.

[1] Department of Immunology and Microbiology, University of Colorado Anschutz Medical Campus, Aurora, CO, USA. [2] Department of Immunology and Genomic Medicine, National Jewish Health, Denver, CO, USA. [3]Present address: Evergrande Center for Immunologic diseases at Harvard Medical School and Brigham and Women's Hospital, Boston, MA, USA. [4]Present address: Stanford Health Care, Department of Pathology, Stanford University, Stanford, CA, USA. ✉email: Laurent.gapin@cuanschutz.edu

Two fundamental features of the adaptive T cell response are its enormous antigen recognition capacity and its ability to form immunological memory. The former endows T cells with a potential repertoire capable of targeting the myriad pathogens a host might encounter while the latter helps the host respond with significantly faster kinetics in the event of a reinfection with a previously encountered pathogen. Although the majority of T cells fits this paradigm, recent work highlights the importance of innate-like T ($T_{inn}$) cells, which differ both in their phenotype and response kinetics. $T_{inn}$ cells, such as invariant natural killer T (iNKT) cells and mucosal-associated invariant T (MAIT) cells, tend to establish permanent residency in tissues, express restricted T cell antigen receptor (TCR) repertoires and share a non-MHC class I or II restriction requirement for antigen recognition[1,2]. Compared to their conventional T cell ($T_{conv}$) counterparts, $T_{inn}$ cells are also not constrained by the need for prior sensitization and exist in a pre-primed "memory" state, ready to respond within minutes of stimulation, despite being antigen-inexperienced[3]. This is due to the fact that unlike $T_{conv}$ cells, $T_{inn}$ cells acquire an effector and memory phenotype over the course of their development in the thymus[4,5].

Thymic development of iNKT cells has been extensively studied over the past decade. Although they share their developmental origins with $T_{conv}$ up to the double positive (DP) thymocyte stage, their paths then diverge[6]. While DP precursors for $T_{conv}$ are selected on peptide-MHC complexes expressed on thymic epithelial cells, iNKT DP precursors are selected on lipids bound to the MHC-I-like molecule (MHC-Ib) CD1d expressed on neighboring DP thymocytes[7]. These DP-DP synapses promote commitment to the iNKT lineage by inducing high levels of the transcription factor Egr2, which in turn leads to expression of the lineage-determining BTB-ZF transcription factor PLZF[8]. PLZF subsequently directs the further differentiation of iNKT cells into three distinct mature effector subsets, iNKT1, iNKT2 and iNKT17, analogous to the CD4[+] T cell polarized subsets observed in the periphery. Additionally, each of these iNKT subsets expresses the canonical master transcription factors that drive its fate, with iNKT1 cells expressing T-bet, iNKT17 expressing Rorγt, and iNKT2 expressing high levels of PLZF and GATA3[4,9]. Thus, iNKT effector subsets arise as a consequence of their thymic development at steady-state instead of differentiation in peripheral tissues due to inflammatory cues. Additionally, some iNKT subsets have been shown to secrete cytokines at steady-state, thereby influencing their surrounding tissue microenvironments in non-redundant manners[9]. More recently, MAIT cells were proposed to follow a similar developmental pathway[10,11], with the key distinguishing feature being that they are selected by other DP cells expressing the MHC-Ib molecule MR1 presenting vitamin metabolites[12]. While a large body of knowledge is available regarding iNKT cell characteristics and development in the thymus and in the periphery[4,5,13–15], the developmental steps underlying iNKT cell differentiation, and by extension MAIT cells, remain incomplete. In particular, prior studies have largely focused on analyzing iNKT cells in bulk, potentially missing developmental intermediate populations.

To overcome this limitation, here, we characterize the transcriptomes of thymic iNKT cells at the single cell level (scRNA-seq) to reveal the underlying heterogeneity that exists over the course of iNKT cell development. Our results illustrate the transcriptional signature of the previously identified iNKT subsets and reveal a largely common developmental path between iNKT and MAIT cells. We further identify their shared requirement for the *Hivep3* gene, with *Hivep3*-deficient mice exhibiting a profound loss of iNKT, MAIT and PLZF[+]Vγ1[+]Vδ6.3[+] γδ T cells. scRNA-seq of *Hivep3*[−/−] thymic iNKT cells highlights a crucial role for *Hivep3* in regulating the expression of several key developmental genes, including *Hdac7*, *Drosha* and *Zbtb16* (the gene coding for PLZF). ATAC-seq analysis of mature iNKT cell subsets shows that in absence of *Hivep3*, chromatin accessibility is increased at regions enriched for NF-κB binding motifs. Altogether, our results identify *Hivep3* as a key regulator that controls, transcriptionally and post-transcriptionally, the development of $T_{inn}$ lymphocytes.

## Results

**Single-cell RNA-seq of thymic C57BL/6 iNKT cells.** We performed droplet-based single cell RNA-seq (scRNA-seq) analysis of PBS57-CD1d-tetramer[+] cells sorted from the thymi of 8-week old female C57BL/6 mice using the Chromium system (10x Genomics). To enrich for early precursors and developmental intermediates, CD1d-tetramer[+] cells were sorted at a 50:50 ratio of CD44[low] to CD44[high] cells. Two independent experiments were performed. The two distinct scRNA-seq datasets were integrated and corrected for batch effects into an integrated reference dataset using the fastMNN algorithm[16]. After quality control steps, a total of 10,506 cells were retained for further analysis. We used uniform manifold approximation and projection (UMAP) for dimensionality reduction to display all iNKT cells in a shared low-dimensional representation (Fig. 1a). To further characterize the subpopulation structures, we applied unsupervised graph-based clustering, which yielded 11 distinct clusters (Fig. 1b), with an average of 1956 genes and 6617 UMI expressed by the cells in each cluster (Fig. 1c). The expression of *Cd24a*, *Cd69*, *Egr2*, *Il2rb*, *Cd44*, *Zbtb16* (encoding PLZF), *Rorc* (encoding Rorγt), *Tbx21* (encoding T-bet), *Gata3* (encoding GATA3), *Ifng*, *Il4*, *Ccr7*, *Ccr6*, *Il17rb* and *Cd4* were used to define immature populations and effector subsets (Fig. 1d). iNKT17 cells (*Rorc*[+], *Ccr6*[+], *Zbtb16*[+], *Cd4*[−]) were found in a single cluster (cluster 5), while iNKT1 cells (*Il2rb*[+], *Tbx21*[+], *Ifng*[+]) were found in five clusters (clusters 6 to 10). iNKT2 cells (*Zbtb16*[+], *Gata3*[+], *Cd4*[+], *Il4*[+]) were contained within 4 clusters (clusters 1 to 4), with three of these clusters (clusters 1, 2 and 3) corresponding to cycling cells (Figs. 1f, 2c, d). The earliest iNKT precursor cells, stage 0 cells, were assigned to cluster 0. Expression of *Cd24a*, the proliferation marker *Ki67*, and the master transcription factors defining each of the mature iNKT cell subsets (*Zbtb16*, *Rorc* and *Tbx21*) confirm these identities (Fig. 1f), with the top five genes that characterize each of these clusters shown in Fig. 1e.

**Transcriptional heterogeneity of thymic iNKT cells.** We next identified the cluster-specific signature genes (Fig. 2a and Supplementary Data 1), performed gene ontology analysis associated with gene expression in each cluster (Fig. 2b) and determined the cell cycle status of the cells (Fig. 2c). We identified 565 stage 0 cells (cluster 0), allowing for a more in-depth analysis of their transcriptional profile than previously achieved[17]. Differentially expressed genes (DEGs) in cluster 0 comprised several genes associated with lymphocyte activation and T cell differentiation (Fig. 2b), including *Itm2a*[18], *Pdcd1* (encoding PD-1), *Cd27*, *Cd28*, *Cd69*, *Slamf6*, *Cd81*, *Ldhb* and *Bcl2* (coding for the anti-apoptotic protein BCL-2) (Fig. 2d). Several genes coding for transcription factors, such as *Lef1* (ref. [19]), *Sox4* (ref. [20]), *Egr2* and *Egr1* (ref. [8]), *Tox*[21], *Id3* (ref. [22]), *Myb*[23], *Ikzf1* and *Ikzf2* (coding for IKAROS and HELIOS, respectively), including some that have been previously implicated in iNKT cell development, were highly expressed in stage 0 iNKT cells (Fig. 2d). High expression of the protein products encoded by some of these genes in stage 0 iNKT cells was verified by flow cytometry, thereby validating the scRNA-seq data (Fig. 2e, f).

A multi-potent CCR7[+] iNKT cell progenitor (iNKTp), with a unique transcriptional signature has been described as a

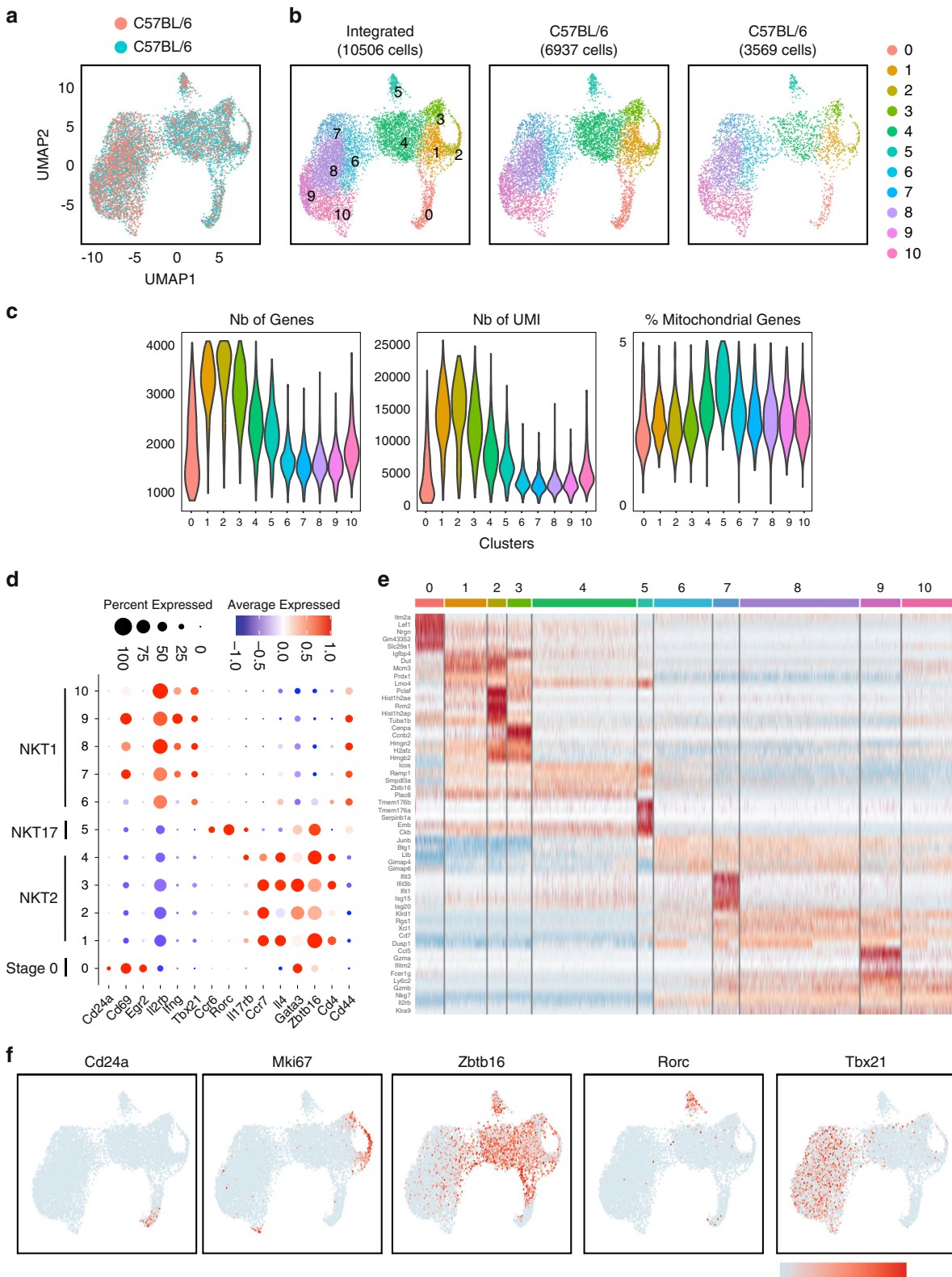

**Fig. 1 scRNA-seq analysis of steady-state C57BL/6 thymic iNKT cells. a** Uniform manifold approximation and projection (UMAP) of two independent scRNA-seq data sets from C57BL/6 thymic iNKT cells integrated using FastMNN and colored by sample of origin. **b** UMAP of 10,506 iNKT cells colored by inferred cluster identity. **c** Violin plot depicting the number of gene, number of UMI and percentage of mitochondrial genes expressed in each cluster. **d** Dot plot showing scaled expression of selected signature genes for each cluster colored by average expression of each gene in each cluster scaled across all clusters. Dot size represents the percentage of cells in each cluster with more than one read of the corresponding gene. **e** Heatmap showing row-scaled expression of the 5 highest differentially expressed genes (DEGs, Bonferroni-corrected *P*-values < 0.05, Wilcoxson-test) per cluster for all iNKT cells. **f** Expression of five typical genes used to define most common iNKT cell subsets and cycling cells.

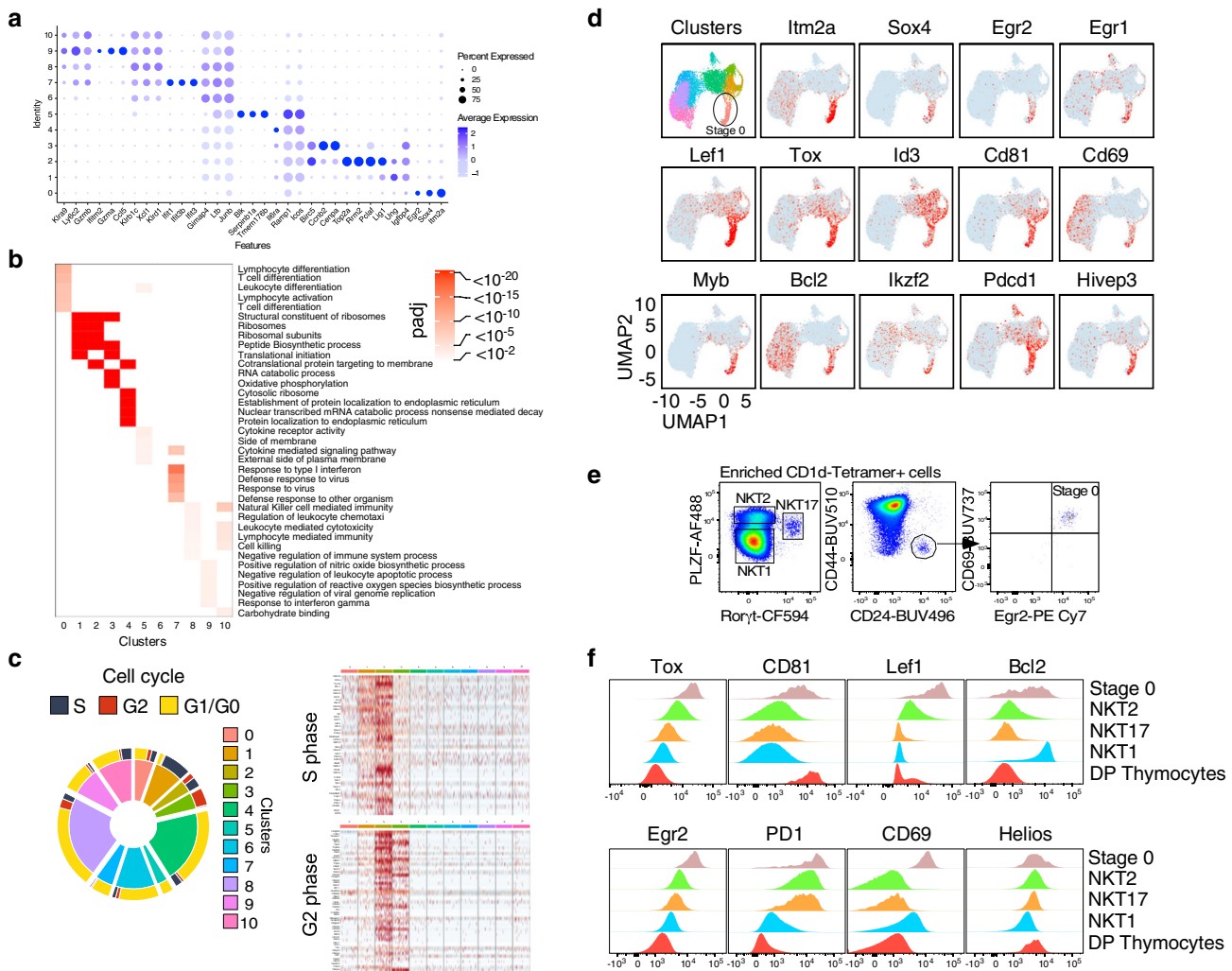

**Fig. 2 Thymic iNKT subsets display distinct gene expression profiles. a** Dot plot showing scaled expression of the top 3 highest DEGs for each cluster colored by average expression of each gene in each cluster scaled across all clusters. Dot size represents the percentage of cells in each cluster with more than one read of the corresponding gene. **b** Gene ontology (GO) analysis of DEGs for each cluster. Selected GO terms with Benjamini–Hochberg-corrected *P*-values < 0.05 (one-sided Fisher's exact test) are shown. Heatmaps display $P_{adj}$ value significance of enrichment of GO terms in each cluster. **c** Pie plot depicting the proportion of cells in each cluster and for each cluster the proportion of cells in each phase of the cell cycle. Heatmap showing row-scaled expression of cell cycle-related genes for each cluster. **d** Expression patterns of fourteen stage 0 iNKT-specific genes in our scRNA-seq data. Each dot represents one cell and gene expression is plotted along a colorimetric gradient, with red corresponding to high expression. **e** Gating strategy to identify stage 0, iNKT1, iNKT2 and iNKT17 cell subsets by flow cytometry. **f** Histograms displaying protein expression levels by flow cytometry within each iNKT subset for the indicated proteins. Protein expression in DP thymocytes is also displayed as a control. Data are representative of at least 2 independent experiments for each marker.

developmental intermediary following commitment to the iNKT cell lineage by stage 0 cells[24]. However, we could not assign this gene signature, composed of 44 different genes, to a transcriptionally unique set of cells and instead observed this gene set expressed by cells in multiple clusters (Clusters 1, 2 and 3) (Fig. 3a). Cells within these clusters have upregulated expression of *Zbtb16* (Fig. 1d–f) as well as several other markers usually associated with the iNKT2 phenotype, including *Icos* and *Izumo1r* (Fig. 3h). These cells have high ribosomal and mitochondrial activity (Fig. 2b), in agreement with their high proliferative status (Fig. 2c). Computation of cell cycle scores revealed that cells in cluster 1 were predominantly in S phase, cells in cluster 2 comprised S and G2 phases, and cells in cluster 3 largely belonged to the G2 phase of the cell cycle (Fig. 2c). These results are in agreement with the proliferative burst known to occur after stage 0 in iNKT cells[25]. Cells in cluster 4 had a similar iNKT2 transcriptional profile (Fig. 3g, h), but with increased

expression of *Il6ra*, *Il17rb*, *Icos*, *Plac8* and *Il4* transcripts, decreased expression of transcripts encoding for *Ccr7*, *Ccr9*, *Il13* and undetectable expression of cell-cycle genes (Fig. 3i, j and Supplementary Data 1).

Cells pertaining to cluster 5 expressed high levels of the master regulator *Rorc* (Fig. 1f) and other genes whose expression have been shown to be directly dependent on Rorγt in $T_H17$ cells[26], including *Tmem176a/b*, *Ltb4r1*, *Il1r1*, and *Il23r* (Fig. 3c and Supplementary Data 1). iNKT17 signature genes also comprised several transcripts whose products are involved in tissue residency (*Itgb7*, *Ccr6*, *Aqp3*) as well as several additional cytokine and cell signaling receptors (*Il17re*, *Il18r1*, *Il7r*, *Sdc1*) (Fig. 3 and Supplementary Data 1) and were defined by their cytokine receptor and signaling pathway activity (Fig. 3).

Cells in the five remaining clusters (clusters 6, 7, 8, 9 and 10) were considered iNKT1 cells, as they expressed transcripts for *Tbx21* (Fig. 1f), *Il2rb* (coding for CD122), *Nkg7*, and *Xcl1*

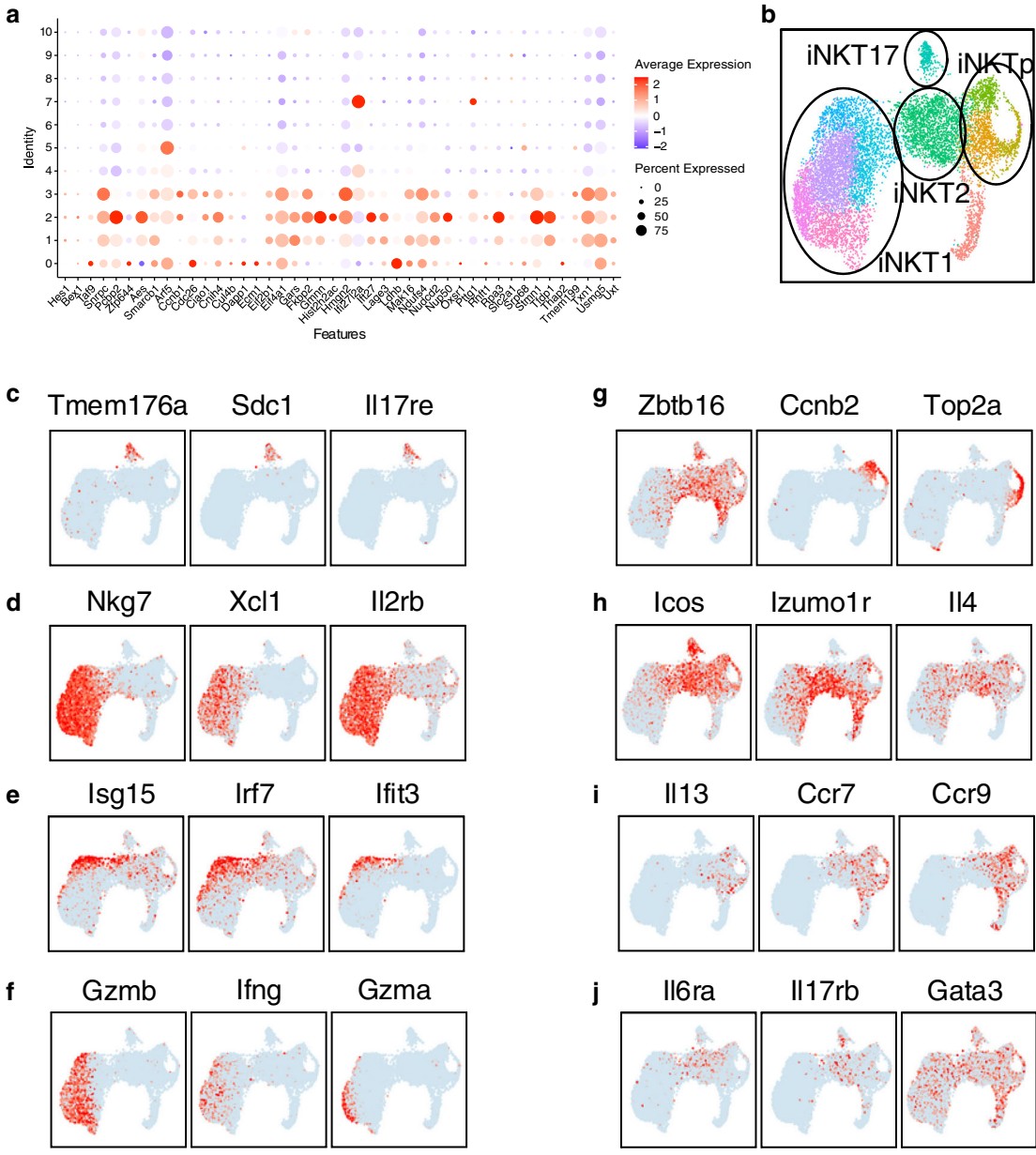

**Fig. 3 Gene expression profile of stage 0 iNKT cells. a** Dot plot showing scaled expression of the 44 genes previously described as the iNKTp signature, for each cluster colored by average expression of each gene in each cluster scaled across all clusters. Dot size represents the percentage of cells in each cluster with more than one read of the corresponding gene. **b** Summary of the UMAP defined in Fig. 1 with major NKT subset populations defined by gates. **c** Expression of three typical genes used to define iNKT17 cells. **d** Expression of three typical genes used to define iNKT1 cells. **e** Expression of three typical genes used to define interferon signaling genes. **f** Expression of three typical genes used to define cells in cluster 9. **g–j** Expression of twelve genes used to illustrate and define the diversity amongst iNKT2/cycling cells.

(Fig. 3d). Interestingly, cells in cluster 6, which neighbor the iNKT2 cells belonging to cluster 4 on the UMAP (Fig. 1), appeared as a transitional population bearing a mixed transcriptional signature observed in cells from cluster 8 (*Klrd1, Xcl1, Cd7, Nkg7, Ms4a4b, Dusp1* and *Dusp2*) and cells from cluster 4 (*Izumo1r*) (Fig. 3h). These cells were nevertheless further defined by higher expression of transcripts encoding for *Junb, Ltb* and several members of the GTPase of the immunity-associated protein (GIMAP) family (*Gimap4, 3 and 6*) (Sup. Data 1). These results suggest that cells in cluster 6 represent an intermediary state between iNKT2 cells and iNKT1 cells. Independent pseudo-temporal ordering using the Monocle v3 (ref. [27]) and Slingshot[28] trajectory inference packages to place iNKT cell populations along possible developmental trajectories, with stage 0 cells defined as the root, supports this possibility (Sup. Fig. 1). Cells in clusters 9 and 10 had enriched expression of transcripts for *Gzma, Gzmb, Ifng, Ccl5, Xcl1, Fcer1g* and *Ly6c2* and killer cell lectin type receptors (*Klrk1, Klre1, Klra5, Klra9, Klrc1, Klrd1, Klrc2*) suggesting that they might represent a further step in the maturation of iNKT1 cells associated with the acquisition of cytotoxicity (Fig. 3f and Supplementary Data 1). Flow cytometry analysis confirmed increase of granzyme A and B proteins expression in Ly6C$^+$ iNKT1 cells compared to Ly6C$^-$ iNKT1 cells (Supplementary Fig. 2). Finally, cells in cluster 7 had a unique transcriptional signature with high expression of type I interferon response genes such as *Isg15* and *20, Ifit1* and *3*,

*Irf7* and *Stat1* (Fig. 3e and Supplementary Data 1), perhaps suggesting that tonic type-I interferon signaling might be involved in the final maturation of iNKT cells, similar to what has been reported for conventional T cell development[29].

**Altered iNKT cell development in *Hivep3*-deficient mice.** We detected high levels of *Hivep3* transcripts in stage 0 iNKT cells (Fig. 2d), which codes for the zinc finger protein HIVEP3 that functions both as a transcription factor and an adapter protein[30–33]. *Hivep3*$^{-/-}$ mice exhibited a large reduction in the proportion and numbers of iNKT cells in all organs examined, including the thymus, spleen, liver, inguinal lymph nodes and lungs (Fig. 4a,b). This represented a cell-intrinsic consequence of *Hivep3* deficiency as revealed by competitive bone marrow chimeras (Fig. 4c). To dissect the developmental stage at which the absence of *Hivep3* expression affected iNKT cells, we performed magnetic bead-based tetramer enrichment of thymic iNKT cells and stained for markers used to define the various stages of development (CD24, CD44 and NK1.1) (Fig. 4d) as well as for transcription factors used to define iNKT cell subsets (PLZF, Rorγt and T-bet) (Fig. 4f). The percentages of cells at stages 0, 1 and 2 were increased in *Hivep3*$^{-/-}$ mice while the proportion of cells at stage 3 was decreased. However, while the total cell number of stage 0 iNKT cells was not significantly different between C57BL/6 and *Hivep3*$^{-/-}$ mice, the number of cells at any subsequent developmental stage was largely and significantly decreased (Fig. 4e). These results indicate that positive selection of iNKT cells is not altered in the absence of *Hivep3* expression, and that the cells are developmentally blocked immediately following stage 0. Surprisingly, the relative proportions of iNKT2, iNKT17 and iNKT1 were only minimally affected, although the numbers of all mature iNKT subsets were dramatically reduced in *Hivep3*$^{-/-}$ thymi (Fig. 4f, g). Thus, the development of iNKT cells in *Hivep3*$^{-/-}$ mice is essentially arrested at stage 0, although a few cells can overcome this block and differentiate into the mature iNKT cell subsets. Nevertheless, we noticed that the remaining iNKT cells in *Hivep3*$^{-/-}$ mice had overall higher TCR expression levels and that the PLZF protein levels at each developmental stage were reduced (Supplementary Fig. 5), collectively hinting at the possibility that the few iNKT remnants in *Hivep3*$^{-/-}$ mice did not develop normally.

**Single cell RNA-seq analysis of Hivep3$^{-/-}$ iNKT cells.** To gain further insight into how *Hivep3* deficiency affects iNKT cell development, we performed two independent scRNA-seq experiments using PBS57-CD1d-tetramer$^+$ cells sorted from the thymi of 8-week old *Hivep3*$^{-/-}$ mice, using a similar sort strategy to the one we used to sort our C57BL/6 samples. The two distinct scRNA-seq datasets were integrated with the C57BL/6 data and corrected for batch effects (Fig. 5a, b). The transcriptional profile of *Hivep3*$^{-/-}$ iNKT cells was rather similar to C57BL/6 iNKT cells, with unsupervised graph-based clustering revealing 10 different clusters common to both genotypes (Fig. 5b). The proportion of cells in cluster 0 (stage 0 cells) was increased in *Hivep3*$^{-/-}$ iNKT cells, in agreement with a block in development at, or immediately following, that stage (Fig. 4). A greater proportion of cells from *Hivep3*$^{-/-}$ than from C57BL/6 mice was also found in cluster 1 (~1% of C57BL/6 cells vs ~7.5% of *Hivep3*$^{-/-}$ cells). Interestingly, these *Hivep3*$^{-/-}$ cells displayed a predominant iNKT1 gene signature (Supplementary Fig. 3). By contrast, the proportion of proliferating cells in clusters 2 and 3 was under-represented in *Hivep3*$^{-/-}$ cells compared to C57BL/6 cells (Fig. 5c), which was confirmed by staining for the Ki67 proliferation marker and BrdU incorporation (Supplementary Fig. 4). Although the proportion of cells in the iNKT2 and

iNKT17 clusters (clusters 4 and 5) did not appear dramatically different between C57BL/6 and *Hivep3*$^{-/-}$ cells, *Hivep3*$^{-/-}$ iNKT1 cells were largely in cluster 9 and fewer cells were assigned to cluster 6 compared to the C57BL/6 iNKT1 cells (Fig. 5b, c). Altogether, these results show that in the absence of *Hivep3*, iNKT development is blocked at stage 0, with a few cells able to further progress in an aberrant fashion, bypassing the proliferative burst and instead acquiring an iNKT1 transcriptional signature earlier during development.

Since *Hivep3* expression in C57BL/6 cells is limited to the early stages of development, we next determined how its absence affected gene expression in stage 0 cells (Supplementary Data 2). We found 65 DEGs between C57BL/6 and *Hivep3*$^{-/-}$ stage 0 cells (Fig. 5d). Eight transcripts were expressed at lower levels in stage 0 *Hivep3*$^{-/-}$ cells compared to C57BL/6 cells - *Cd52*, *Dusp1*, *Zbtb16*, *Plac8*, *Id2*, *Ccr9* and *Drosha* (Fig. 5d, e). All other stage 0 DEGs had higher expression in *Hivep3*$^{-/-}$ cells compared to C57BL/6 cells, including transcripts encoding the TCR chains and the CD3ζ subunit (*Trbc2*, *Trbc1*, *Trac*, *Cd247*) (Fig. 5f), potentially explaining the higher level of TCR expression observed on *Hivep3*$^{-/-}$ cells. Transcripts for several TFs downstream of TCR signaling, including *Egr1*, *Egr2*, *Nr4a1*, *Id3*, *Ikzf3* (Fig. 5f), also displayed higher expression in *Hivep3*$^{-/-}$ cells, suggesting that the KO cells might have received stronger signaling during positive selection. Interestingly, *Hivep3*$^{-/-}$ stage 0 iNKT cells also expressed higher levels of transcripts encoding histone deacetylase 7 (*Hdac7*, Fig. 5f), which was recently shown in a gain-of-function mouse transgenic model to similarly interfere with iNKT cell development[34].

Analyzing the remaining 9 clusters across strains revealed 785 DEGs between C57BL/6 and *Hivep3*$^{-/-}$ cells (Supplementary Data 2). Notable amongst these DEGs were the decreased levels of *Ccr9* and *Ccr7* transcripts in *Hivep3*$^{-/-}$ cells compared to C57BL/6 cells, whose protein products are required for the migration of iNKT cell precursors from the cortex to the medulla[35]. Transcripts encoding two Bcl-2 family members, *Bcl2a1b* and *Bcl2a1d*, were also largely decreased in *Hivep3*$^{-/-}$ cycling cells, perhaps affecting their survival (Fig. 6 and Supplementary Data 2). *Cd4* and *Icos* transcripts were also decreased (Fig. 6), which was further reflected at the protein level (Supplementary Fig. 5). Both *Il13* transcripts, which are found in early cycling iNKTp cells, and *Il4* transcripts, which are otherwise expressed in cycling and more mature iNKT2 cells, were reduced in the absence of *Hivep3* (Fig. 6). These results suggest that *Hivep3*$^{-/-}$ iNKT cells do not express appropriate amounts of the pre-formed mRNAs encoding these two cytokines, a hallmark of iNKT cell differentiation[36,37]. Perhaps as a consequence, stimulation of peripheral iNKT cells in vivo with the agonist ligand α-GalCer, showed a defect in IL-4 production, while IFNγ production was less affected (Supplementary Fig. 6). Similar to what was observed in progenitor cells (Fig. 5e), the levels of *Zbtb16* transcripts were also found significantly decreased in subsequent developmental stages lacking *Hivep3* expression (Fig. 6 and Supplementary Fig. 5). Finally, expression of transcripts encoding the ribonuclease DROSHA, which is responsible for the initiation step of miRNA processing, was mostly absent in *Hivep3*$^{-/-}$ cells (Fig. 6), perhaps contributing to the observation that 70% of DEGs in these 9 clusters display higher expression levels in *Hivep3*$^{-/-}$ cells compared to C57BL/6 cells. These included several iNKT1-associated transcripts, such as *Klra1* and *Klra5* (Fig. 6 and Supplementary Data 2).

**HIVEP3 activity influences chromatin landscape of iNKT subsets.** To assess whether and how HIVEP3 protein might influence iNKT cell development via regulatory elements,

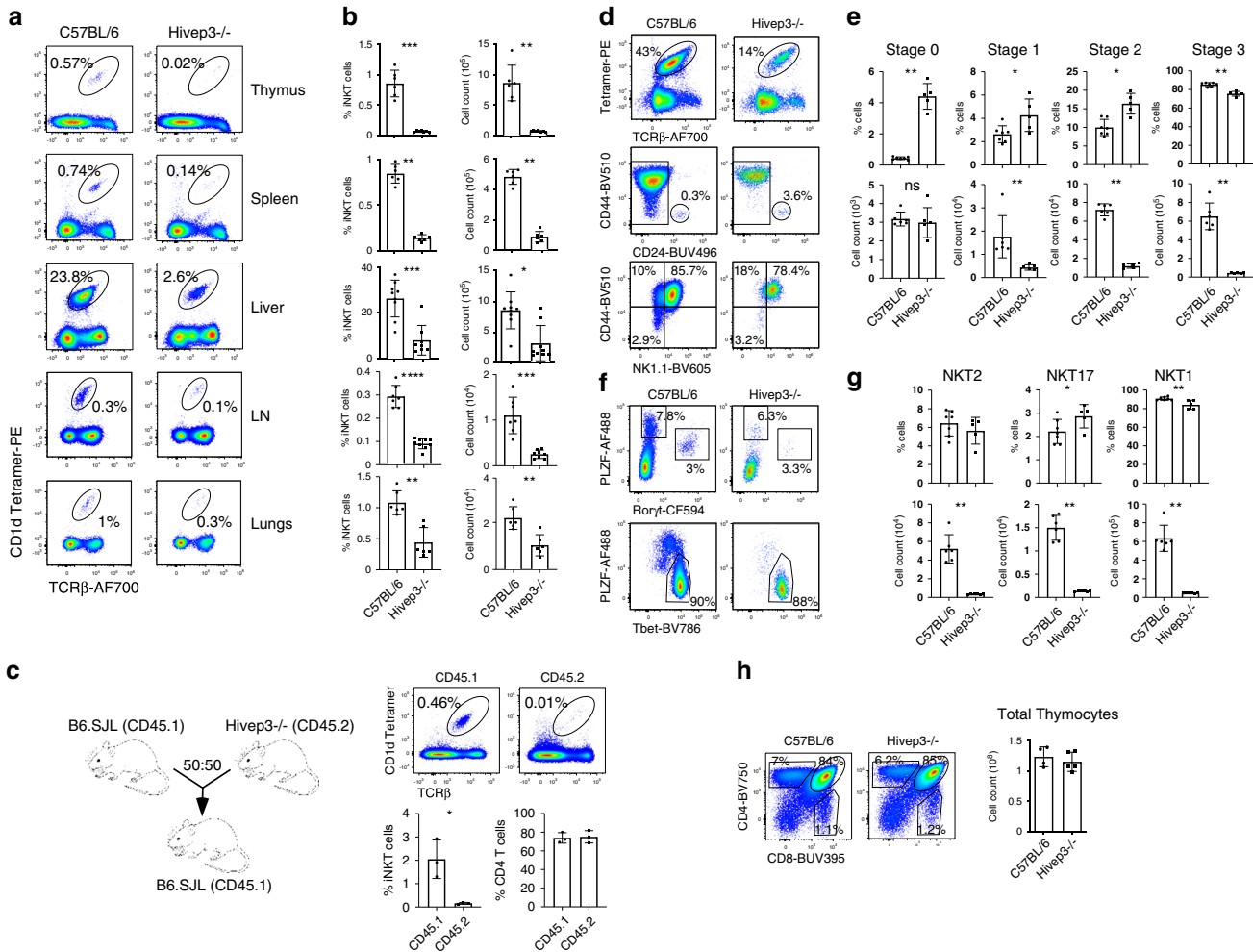

**Fig. 4 Intrinsic deficiency of iNKT cells in *Hivep3$^{-/-}$* mice. a** iNKT cell proportions in C57BL/6 and *Hivep3$^{-/-}$* mice were identified in the indicated tissues by staining with PBS57-CD1d tetramers and an antibody targeting TCRβ. For the lung and liver samples, cells were first gated on viability dye negative, CD45$^+$ and CD19$^-$ cells, while for the other organs cells were gated on viability dye negative cells. **b** Quantified iNKT cell proportions and numbers in C57BL/6 and *Hivep3$^{-/-}$* mice in each of the indicated tissues. Data are from 3 independent experiments with 6 thymus, 6 spleen, 8 liver, 7 lymph node and 6 lung tissues analyzed from C57BL/6 mice and 7 thymus, 6 spleen, 9 liver, 8 lymph node and 7 lung tissues analyzed from *Hivep3$^{-/-}$* mice. Thymus $p = 0.0006$ and $p = 0.0012$, Spleen $p = 0.022$ and $p = 0.022$, Liver $p = 0.0003$ and $p = 0.0111$, LN $p < 0.0001$ and $p = 0.002$, Lungs $p = 0.022$ and $p = 0.043$ for percentages and absolute numbers, respectively. **c** *Left*, Cartoon depicting the competitive bone marrow chimera strategy in which CD45.1 C57BL/6 bone marrow was mixed at a 50:50 ratio with CD45.2 *Hivep3$^{-/-}$* bone marrow and subsequently transferred into irradiated CD45.1 recipient mice. *Right*, iNKT and single positive CD4$^+$ T cell proportions were quantified from both C57BL/6 (CD45.1) and *Hivep3$^{-/-}$* (CD45.2) mice after immune reconstitution. Data are from one experiment with 3 mice per condition. **d–g** Data are from 2 independent experiments with a total of 7 C57BL/6 and 5 Hivep3$^{-/-}$ mice. iNKT cells from C57BL/6 and *Hivep3$^{-/-}$* mice were categorized based on expression of CD24, CD44, NK1.1 into stage 0, stage 1, stage 2 and stage 3 and subsequently quantified for their relative proportions and numbers. Stage 0 $p = 0.0025$ and $p = 0.9$, Stage 1 $p = 0.048$ and $p = 0.0027$, Stage 2 $p = 0.01$ and $p = 0.0025$, Stage 3 $p = 0.0025$ and $p = 0.0023$ for percentages and absolute numbers, respectively (**d**, **e**) and also categorized based on expression of PLZF, Rorγt and T-bet into iNKT1, iNKT2 and iNKT17 and subsequently quantified for their relative proportions and numbers iNKT1 $p = 0.0025$ and $p = 0.0025$, iNKT2 $p = 0.3434$ and $p = 0.0027$, iNKT17 $p = 0.046$ and $p = 0.0025$ for percentages and absolute numbers, respectively (**f**, **g**). **h** *Left*, Flow cytometry analysis depicting DP, CD4 and CD8 proportions in WT and KO mice. *Right*, Histograms quantifying total thymic cellularity. Data are from 2 independent experiments with a total of 4 C57BL/6 and 5 Hivep3$^{-/-}$ mice. Data in the figure are mean ± SD with dots representing individual values. Source data are provided as a source data file.

we compared chromatin accessibility between C57BL/6 and *Hivep3$^{-/-}$* iNKT1, iNKT2 and iNKT17 cells by ATAC sequencing (ATAC-seq). iNKT1, iNKT2 and iNKT17 cell subsets each showed a unique chromatin landscape, with regions of accessibility specific to each subset (Supplementary Fig. 7). The analyses showed that *Hivep3$^{-/-}$* samples were closely related to the C57BL/6 samples for each iNKT cell subset, based on global chromatin accessibility signal compared by pairwise Euclidean distances between each individual replicate (Fig. 7a). To investigate whether there were specific changes in chromatin

accessibility in iNKT cells from *Hivep3$^{-/-}$* mice, we compared the accessible regions between C57BL/6 and *Hivep3$^{-/-}$* backgrounds for each subset (Fig. 7b). We identified 498 regions in iNKT1 cells and 730 regions in iNKT2 that were significantly (3-fold change, $p_{adj} < 0.1$) differentially accessible between C57BL/6 and *Hivep3$^{-/-}$* cells. In contrast, only 102 regions were found differentially accessible between C57BL/6 and *Hivep3$^{-/-}$* iNKT17 cells. Some, but not all, of these differentially accessible regions surrounded genes that were also found differentially expressed transcriptionally. For example, the chromatin surrounding the

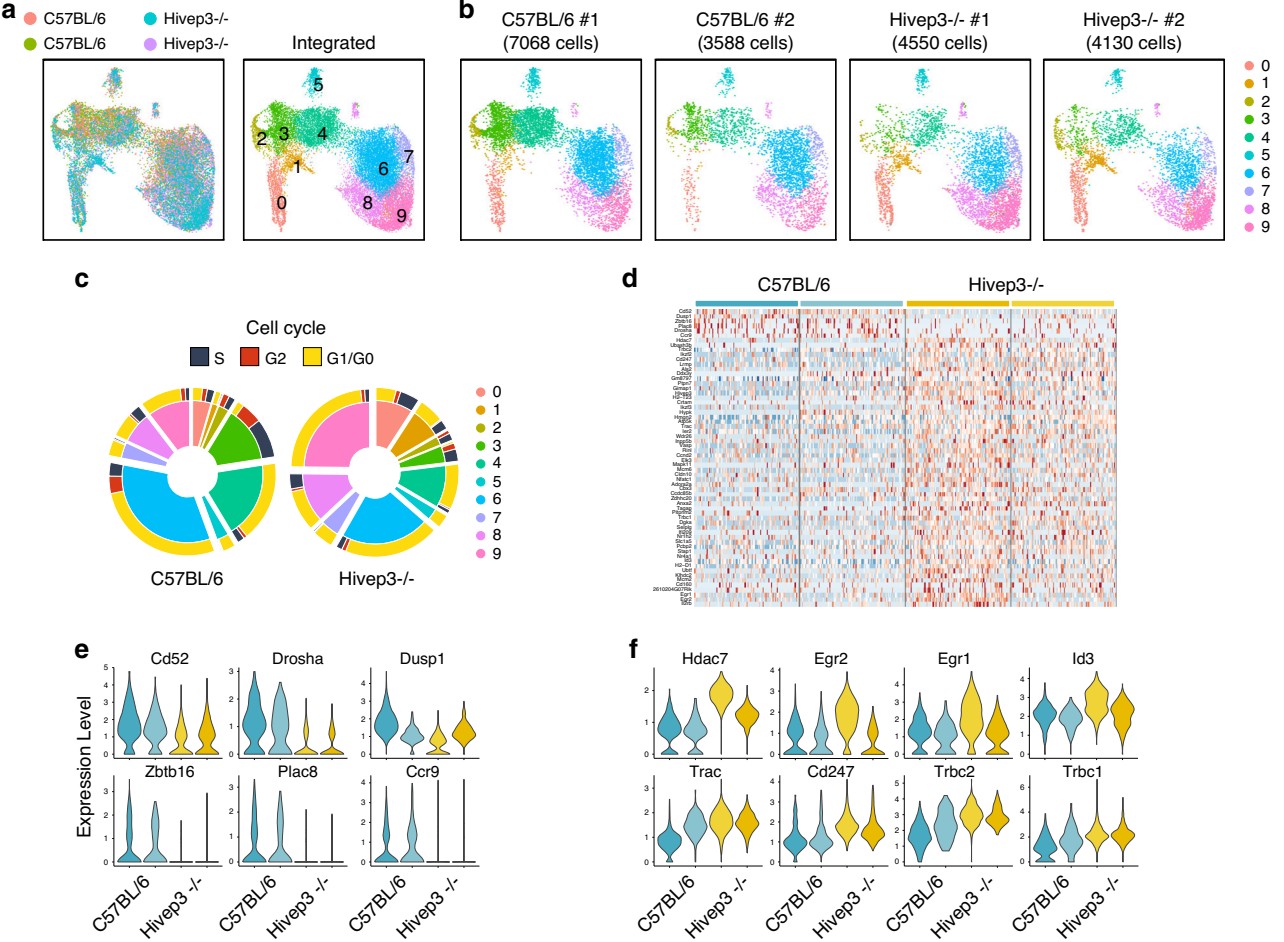

**Fig. 5 scRNA-seq analysis of steady-state thymic iNKT cells from Hivep3⁻/⁻ mice. a** *Left*, Uniform manifold approximation and projection (UMAP) displaying two independent scRNA-seq data sets from C57BL/6 and two independent *Hivep3⁻/⁻* thymic iNKT cells colored by sample of origin after MNN batch-correction. *Right*, UMAP colored by inferred cluster identity for the integrated dataset. **b** Individual UMAP plots for each biological sample in the dataset, with the sample and number of cells passing quality control and filtering indicated above each plot. **c** Cell cycle scores were computed for each cell in the dataset using the R Seurat package and then categorized into S, G2 or G1/G0 phases. Pie charts depict the proportion of C57BL/6 (*Left*) and *Hivep3⁻/⁻* (*Right*) cells within each cluster belonging to a given phase of the cell cycle. **d** Heatmap displaying expression levels of 65 differentially expressed genes across stage 0 iNKT cells in the dataset. The sample to which a given cell belongs are indicated at the top of the heatmap. **e**, **f** Violin plots displaying aggregate expression levels per sample of genes that are expressed higher in C57BL/6 (**e**) or higher in *Hivep3⁻/⁻* (**f**) stage 0 iNKT cells.

*Cd247* gene was clearly more accessible in *Hivep3⁻/⁻* cells compared to C57BL/6 cells, irrespective of subset (Fig. 7c), in agreement with higher *Cd247* transcript levels in *Hivep3⁻/⁻* cells (Sup. Data 2). By contrast, accessibility of peaks surrounding the *Ccr9* and *Drosha* genes was diminished in *Hivep3⁻/⁻* cells compared to C57BL/6 cells and this was most apparent in specific iNKT cell subsets (Fig. 7c). To associate changes in chromatin accessibility between different iNKT subsets with potential transcriptional regulators, we compared the changes in ATAC-seq signal at peaks containing known TF binding motifs between all samples using the ChromVAR package (Fig. 7d). As previously shown[38], iNKT1 cells had the highest signal at regions containing Tbox, Runt and Ets binding motifs; peaks containing TCF, NFAT and Egr binding motifs had highest signal in iNKT2 cells, and peaks containing Rorγt binding sites had the highest signal in iNKT17 cells (Fig. 7d). Interestingly, the analysis also demonstrated that peaks containing the NF-κB binding motif were more accessible in *Hivep3⁻/⁻* cells compared with C57BL/6 cells, in all three subsets (Fig. 7d). This is in agreement with previous reports showing that HIVEP3 can directly compete with NF-κB for DNA binding[39]. Finally, *Hivep3⁻/⁻* iNKT1 cells displayed an increased

in accessibility in Runt and bZIP motifs-containing peaks (Fig. 7d), perhaps predisposing them to a more terminal iNKT1 phenotype. Altogether, these results are consistent with a role for HIVEP3 in affecting chromatin accessibility and thereby modulating the expression of certain genes. However, not all differentially expressed genes had corresponding changes in chromatin accessibility, suggesting that the effects of HIVEP3 on iNKT cell differentiation may be linked to its activity as an adapter protein or due to indirect, or secondary, transcriptional effects.

**PLZF⁺ innate T cells in *Hivep3*-deficient mice.** PLZF expression is necessary and sufficient for many of the salient features that characterize $T_{inn}$ cell function and phenotype. Therefore, we wondered whether *Hivep3* might also be expressed during the development of other $T_{inn}$ cells and whether its absence would similarly affect their development. The transcriptional heterogeneity and developmental paths of thymic MAIT cells were recently examined by scRNA-seq using the Chromium platform[10]. We re-analyzed this MAIT dataset and integrated the data with our iNKT datasets (Fig. 8a, b). Both iNKT and MAIT cells distributed similarly on the UMAP and populated all clusters

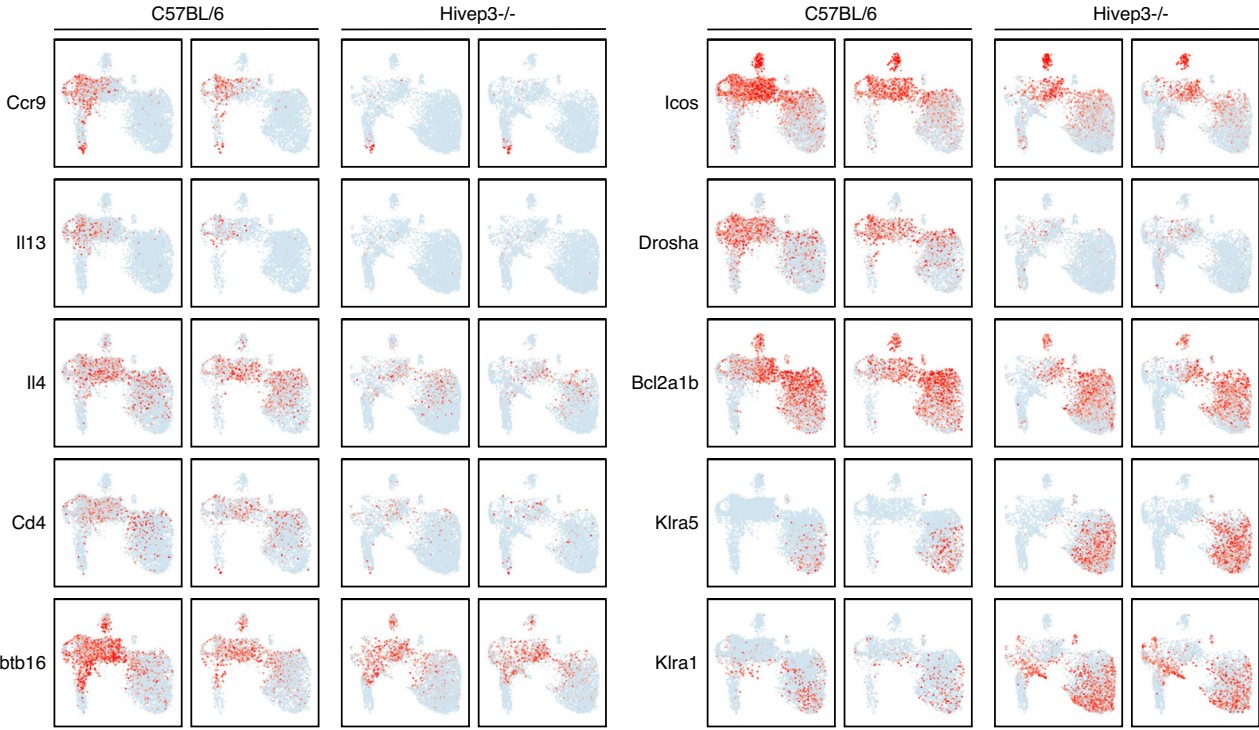

**Fig. 6 Expression profiles of genes differentially expressed between C57BL/6 and *Hivep3*$^{-/-}$ iNKT cells by scRNA-seq.** Expression of selected genes in the C57BL/6 and *Hivep3*$^{-/-}$ single cell RNA-seq samples. Each dot represents one cell and gene expression is plotted along a colorimetric gradient, with red corresponding to high expression.

(Fig. 8b). MAIT cells were found predominantly in a cluster of cells with a $T_H17$-like transcriptome (cluster 6), while iNKT cells mainly occupied clusters of cells with a $T_H1$-like transcriptome (clusters 7, 8, 9, 10 and 11) (Fig. 8b). These results are in agreement with the expected frequencies for each of these subsets in both populations[40], and demonstrate that a similar transcriptional profile is shared between these two thymic $T_{inn}$ populations. It also suggests that both cell types follow a conserved developmental program, supported by the greater similarities in the gene expression profiles between iNKT and MAIT cells belonging to a given cluster than to cells in other clusters (Supplementary Fig. 8). Cluster-specific signature genes common to both cell types outlined the genes that define this core developmental program (Supplementary Data 3). Early CD24$^+$ MAIT cells progenitors (in cluster 0) share expression of many genes with Stage 0 iNKT cells, including *Itm2a*, *Cd24a* and *Hivep3* (Fig. 8c, d and Supplementary Data 3). In mice, the proportion and total number of MAIT cells is much lower than for iNKT cells, making their detection more challenging. Nevertheless, in our mouse colony, we can readily detect MAIT cells in the inguinal lymph nodes and lungs using MR1 tetramers loaded with the agonist antigen 5-OP-RU (Fig. 8e). *Hivep3*$^{-/-}$ mice exhibited a large reduction in the proportion and numbers of MAIT cells in these organs (Fig. 8e, f). Altogether, these results demonstrate that early in the developmental path of MAIT cells, HIVEP3 is expressed and that in its absence, the proportion and numbers of MAIT cells are strongly diminished.

A subset of mouse γδ T cells also express high levels of PLZF[41,42]. These Vγ1$^+$Vδ6.3$^+$ T cells have the capacity to rapidly secrete cytokines and chemokines upon stimulation, a functionality that is dependent on PLZF expression[43]. We found a significant decrease in the proportion of Vγ1$^+$Vδ6.3$^+$ T cells amongst total γδ T cells in the thymus, spleen, lymph nodes and lungs of *Hivep3*$^{-/-}$ mice compared to C57BL/6 mice (Fig. 8g), with the *Hivep3*$^{-/-}$ Vγ1$^+$Vδ6.3$^+$ T cells expressing low levels of

PLZF (Fig. 8g, h). We similarly observed a large decrease in a population of αβ T cells that was neither iNKT nor MAIT cells, and yet co-expressed PLZF and Rorγt, in the inguinal lymph nodes of *Hivep3*$^{-/-}$ mice compared to C57BL/6 mice (Supplementary Fig 9). These cells, which are predominantly of the CD4$^-$ CD8$^-$ phenotype, are present in the inguinal lymph nodes of CD1d1d2$^{-/-}$ mice, indicating that they do not correspond to type II NKT cells. Interestingly, innate lymphoid cells (ILCs), which also require PLZF expression early in their development[44], were detected in normal proportions and numbers in the lungs of *Hivep3*-deficient mice (Supplementary Fig 10a, b). Additionally, Foxp3$^+$ regulatory T cells[45] and the precursors of intraepithelial lymphocytes[46], two PLZF-independent T cell populations known to require "agonist" selection[47], just as $T_{inn}$ cells do, were found at comparable frequencies and numbers between C57BL/6 and *Hivep3*$^{-/-}$ mice (Supplementary Fig 10c–f). Altogether, our results uncovered an important role for HIVEP3 in uniquely controlling the development of PLZF$^+$ $T_{inn}$ cells.

## Discussion

Exhibiting characteristics of both innate and adaptive immunity, innate-like T lymphocytes such as iNKT cells, MAIT cells, and subsets of γδ T cells, have emerged as key players in the control of immunity and tissue homeostasis[48–51]. However, a complete mechanistic understanding of $T_{inn}$ differentiation and function remains elusive. We used scRNA-seq to investigate the transcriptional landscape of iNKT cell maturation and fate decisions under steady-state conditions. Unsupervised clustering identified cells with transcriptional signatures of the previously identified stage 0, iNKT1, iNKT2 and iNKT17 cells. iNKT17 cells were transcriptionally homogenous, forming only one cluster in our analysis. However, iNKT2 and especially iNKT1 cells displayed unexpected heterogeneity, revealing the existence of transitional stages that accompany the acquisition of their effector programs.

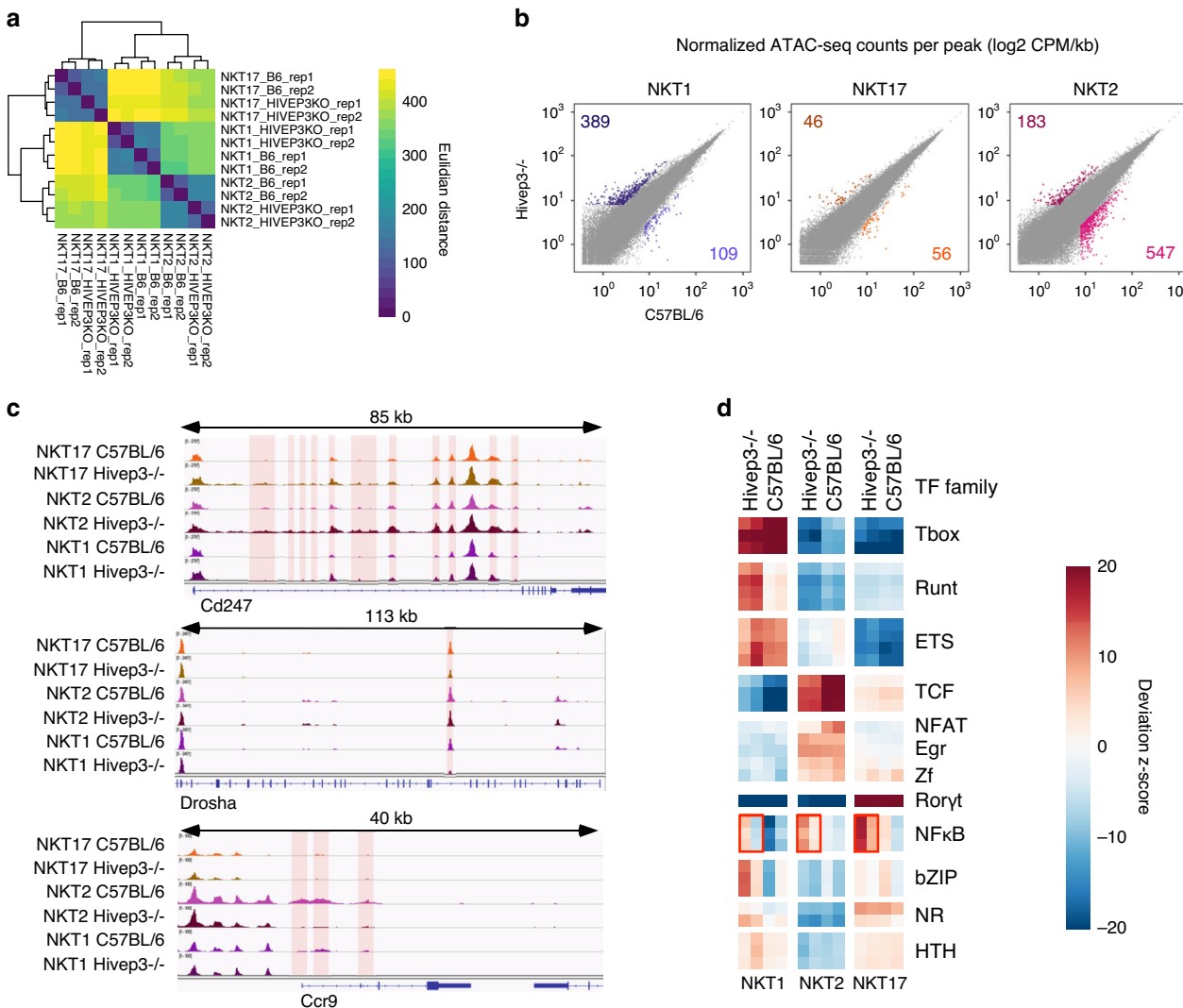

**Fig. 7 *Hivep3* deficiency impacts the chromatin landscapes of thymic iNKT cell subsets. a** Euclidean distance values were computed for the iNKT1, iNKT2 and iNKT17 ATAC-seq samples from both C57BL/6 and *Hivep3*[−/−] mice. **b** Scatterplots of mean ATAC-seq counts per peak comparing iNKT1, iNKT2 and iNKT17 subsets between C57BL/6 and *Hivep3*[−/−] mice. **c** Mean ATAC-seq coverage at the *Cd247*, *Drosha* and *Ccr9* loci in all samples is depicted. Differentially accessible loci are highlighted in red. **d** Motif enrichment for iNKT1, iNKT2 and iNKT17 subsets across both strains within the differentially accessible regions is depicted as a heatmap. NF-κB binding motifs are enriched in the differentially accessible regions in the *Hivep3*[−/−] samples and this is indicated with red boxes.

Stage 0 iNKT cells represented the earliest precursor cells, having just committed to the iNKT cell lineage and expressed several transcripts associated with T cell activation. These cells were adjacent to three clusters of cells defined by high cell-cycle scores, which was in agreement with the intra-thymic proliferative expansion that occurs post-stage 0 iNKT cells[6,25]. Interestingly, these cells were also enriched for a transcriptional signature previously ascribed to intermediary precursor cells, iNKTp cells[24], and were defined by the expression of *Ccr7*, *Ccr9* and *Il13*-encoding transcripts, corroborating previous fate-mapping experiments that had demonstrated that IL-13-expressing intermediate cells can give rise to all mature iNKT subsets[37]. These cells were also lacking *Cd44* transcripts echoing the previously defined stage 1 iNKT cells. Adjacent to the iNKTp cells on the UMAP were cells that were not actively cycling and had acquired a *bona-fide* iNKT2 transcriptional program (*Zbtb16*[+], *Icos*[+], *Izumo1r*[+], *Il6ra*[+], *Il4*[+] but *Ccr7*, *Ccr9* and *Il13* negative). Surprisingly, within this cluster of iNKT2 cells, we detected cells in "bridge" regions of the UMAP with

transcriptional signatures straddling those of iNKT1 and iNKT17 cells. This suggests that iNKT2 cells might not solely represent a terminally differentiated subset of iNKT cells[9], but instead also include cells undergoing further differentiation towards an iNKT17 or iNKT1 subset. Such an interpretation was supported by pseudo-temporal ordering as well as other recently published iNKT scRNA-seq data[52]. Although this conflicts with previous intra-thymic transfer experiments showing that IL-4-secreting iNKT2 cells maintain their phenotype 4 days post-transfer[9], future scRNA-seq experiments coupled with fate-mapping studies of IL-4-producing iNKT2 cells should help further refine our understanding of the developmental path undertaken by thymic iNKT cells. Unexpectedly, we also observed transcriptional diversity amongst iNKT1 cells, suggesting a progressive maturation process in which, concomitant with the detection of transcripts encoding Ly6C and killer cell lectin type receptors, cells gained expression of *Ifng* transcripts and cytotoxic molecules (*Gzmb*, *Gzma*). Interestingly, thymic iNKT1 cells expressed *Cd69* transcripts and protein, suggesting that they are receiving

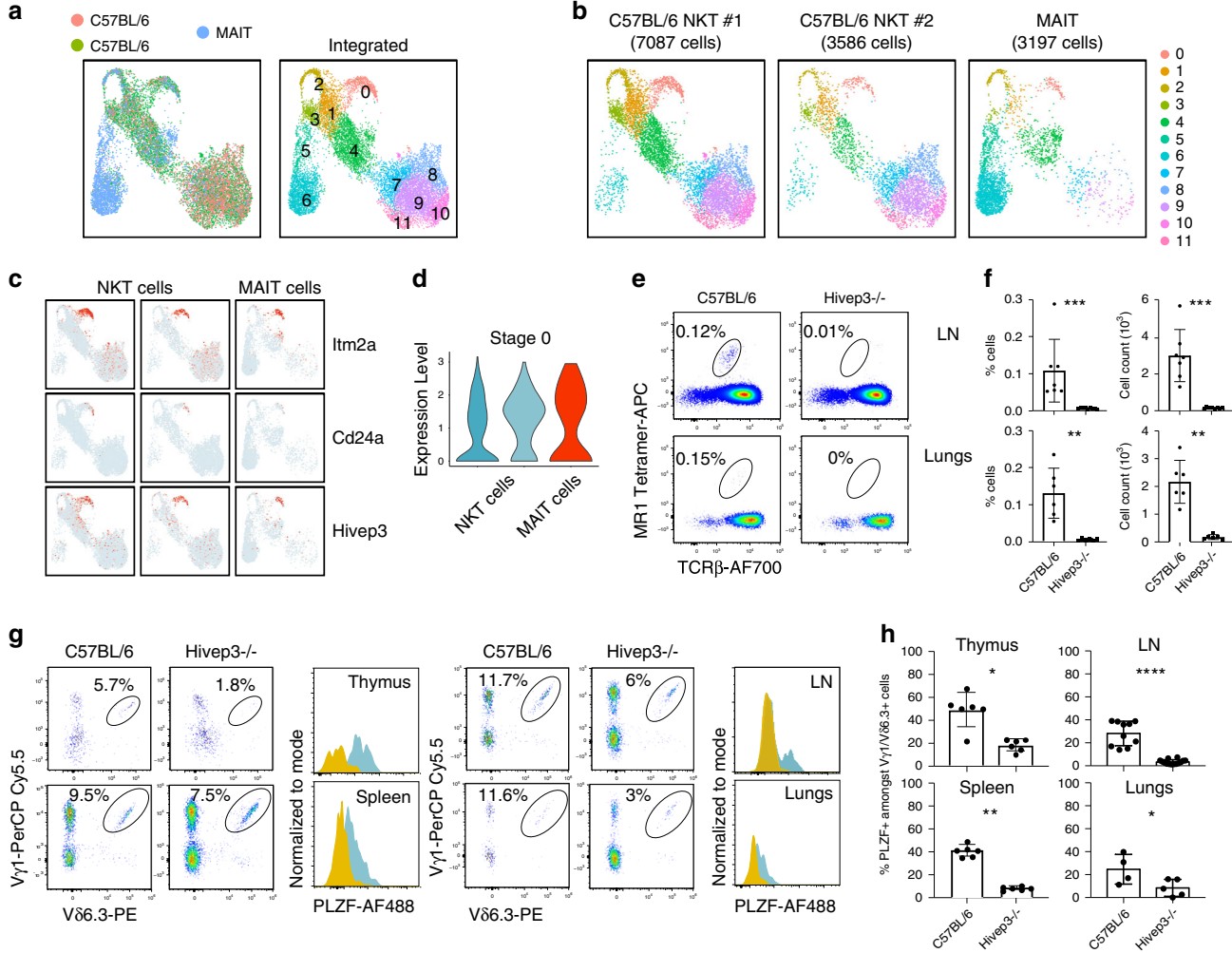

**Fig. 8 *Hivep3* deficiency universally impacts PLZF-expressing innate-like T lymphocytes. a** *Left*, UMAP displaying two iNKT and one MAIT scRNA-seq sample colored by sample of origin after MNN batch-correction. *Right*, UMAP colored by inferred cluster identity for the integrated dataset. **b** Individual UMAP plots for each biological sample in the dataset, with the sample and number of cells passing quality control and filtering indicated above each plot. **c** Expression of selected genes in the iNKT and MAIT scRNA-seq samples. Each dot represents one cell and gene expression is plotted along a colorimetric gradient, with red corresponding to high expression. **d** Violin plots displaying aggregate expression levels of *Hivep3* in stage 0 cells for each sample. **e** MAIT cells in C57BL/6 and *Hivep3*$^{-/-}$ mice were identified in the indicated tissues by staining with the MR1 tetramer and an antibody targeting TCRβ. **f** Quantified MAIT cell proportions and numbers in C57BL/6 and *Hivep3*$^{-/-}$ mice in each of the indicated tissues. Data are from two independent experiments with a total of 5-7 mice per group. LN: $p = 0.0002$ and $p = 0.0022$, Lungs: $p = 0.0022$ and $p = 0.0022$ for percentages and absolute numbers, respectively. **g** Vγ1$^+$ Vδ6.3$^+$ γδ T cells in C57BL/6 and *Hivep3*$^{-/-}$ mice were identified in the indicated tissues by staining with antibodies targeting Vγ1 and Vδ6.3. PLZF expression in these cells from each tissue is plotted as an overlaid histogram with C57BL/6 cells colored in blue and *Hivep3*$^{-/-}$ cells colored in yellow. **h** Quantified PLZF$^+$ Vγ1$^+$ Vδ6.3$^+$ cell proportions in C57BL/6 and *Hivep3*$^{-/-}$ mice in each of the indicated tissues. Thymus $p = 0.0108$, LN $p = 0.0002$, Spleen $p = 0.0022$, Lungs $p = 0.0022$. Data are from 2–3 independent experiments with a total of 6-7 total mice per group. All the data are mean ± SD with dots representing individual values. Source data are provided as a source data file.

activating signals. However, since we did not find further tran-scriptional evidence for TCR-mediated signaling in iNKT1 cells, we suspect that CD69 upregulation might be induced through cytokine signaling instead. For example, IL-15 and type I inter-feron have been previously shown to upregulate CD69 in cells sensitive to these cytokines[53,54]. Along these lines, a clear cluster of iNKT1 cells with type I interferon gene response signature was also identified, suggesting a potential role for type I interferon signaling in the functional maturation of iNKT1 cells.

Aggregation of the scRNA-seq data from thymic iNKT and MAIT cells revealed that both cell types populated the same clusters, indicating that both cell types share their transcriptional profiles and follow similar developmental paths, in agreement with recent findings[55]. While the existence of IL-4-secreting thymic MAIT2 cells remains debatable[55], the data clearly showed

the existence of a few MAIT cells with a transcriptional profile similar to iNKT2 cells (cluster 4, Fig. 8). In agreement with the notion that these cells represent a transitory developmental stage, such MAIT2-like cells might not be licensed to secrete IL-4, like the comparable population of iNKT2 cells which secrete IL-4 upon encountering CD1d-expressing APCs, as recently shown[56]. Such an interpretation would suggest that IL-4 secretion may not represent a terminal differentiation step and that IL-4$^+$ cells are otherwise transcriptionally very similar to non-IL-4 secreting cells. In agreement with this hypothesis, scRNA-seq analysis of IL-4-secreting iNKT2 cells (CD122$^{neg}$, ICOS$^{pos}$, CD138$^{neg}$) and non-IL4-secreting iNKT2 cells sorted from the thymus of KN2 IL4-reporter mice[57] revealed that cells in both samples are tran-scriptionally very similar and populate the same clusters (Sup-plementary Fig 11). However, the IL-4 negative population

contains more cells that appear developmentally poised to become iNKT1 or iNKT17 cells compared to IL-4 positive iNKT2 cells. It is also possible that in contrast to thymic iNKT2 cells, thymic MAIT2-like cells do not encounter medullary MR1-expressing APCs that can trigger their expression of IL-4.

We identified the zinc-finger HIVEP3 transcription factor as exclusively necessary for the accumulation of $T_{inn}$ cells expressing PLZF. In iNKT cells, the absence of HIVEP3 affected the expression of multiple genes. For some of these genes, HIVEP3 likely disturbed their expression by modifying chromatin accessibility of regulatory regions. For example, chromatin accessibility surrounding the Ccr9 gene locus, was noticeably affected by the absence of Hivep3 expression, correlating with gene and protein expression. We also observed changes in gene expression that were not associated with any apparent change in chromatin accessibility. In such cases, modulation of gene expression might be mediated through the adapter activity of HIVEP3. The mechanistic and possible binding partners of HIVEP3 in iNKT cells remain to be explored. Our results also revealed a near absence of transcripts encoding the microRNA processing enzyme DROSHA in Hivep3-deficient iNKT cells. In contrast with $T_{conv}$, $T_{inn}$ cells are particularly sensitive to disruption of miRNA function, both globally and on the individual miRNA level[58–61]. We speculate that with limited Drosha expression, miRNA processing might have been affected and contributed to the dysregulation of iNKT development. In support of this hypothesis, we observed that a large fraction of DEGs showed higher levels of expression in Hivep3-deficient cells compared to C57BL/6 cells, perhaps reflecting a deficiency in post-transcriptional regulation of gene expression. Future experiments analyzing miRNAs expression in Hivep3-deficient iNKT cells will explore this possibility.

The class IIa histone deacetylase HDAC7 is highly expressed in double-positive thymocytes and is localized in their nuclei[62]. Upon thymic selection, HDAC7 is exported out of the nucleus upon TCR stimulation, which thereby affects the transcriptional program of developing T cells[63]. We observed an increase in Hdac7 transcript levels in stage 0 iNKT cells of $Hivep3^{-/-}$ mice compared to their wildtype counterparts. Although the localization of HDAC7 in these cells is currently unknown, it is noteworthy that transgenic mice with a T-cell specific and transient expression of a mutant HDAC7 lacking the phosphorylation sites required for nuclear export, exhibit a dramatic loss of iNKT cells due to reduced proliferation and transcriptional repression of PLZF expression[34]. These results resonate with our observations in $Hivep3^{-/-}$ mice showing a block in iNKT cell development with reduced thymic expansion and decreased PLZF expression. Reduced PLZF protein expression early in the course of iNKT development has been associated previously with significant reduction in iNKT cell numbers[64,65].

$Hivep3^{-/-}$ mice had a profound defect in MAIT cells together with other populations of $T_{inn}$ cells, including γδ NKT cells. Expression of Hivep3 transcripts was recently noted in some γδ progenitors although its role in γδ T cell development was not investigated[66]. Innate lymphoid cells, which also require PLZF for their development, were not affected by the absence of Hivep3 expression, suggesting that HIVEP3 requirement does not extend to all PLZF+ immune cells but may be restricted to TCR expressing cells. This would be consistent with the induction of Hivep3 expression in T cells upon TCR engagement[31], and upon positive selection during their development[67]. However, we did not detect obvious deficiencies in $T_{conv}$ or in other "agonist"-selected T cells, such as $T_{regs}$ and IELPs in $Hivep3^{-/-}$ mice. Like iNKT and MAIT cells, these agonist selected cells require strong agonist TCR signaling during selection. However, they do not need PLZF expression for proper development and are not

selected by neighboring MHC-expressing DP thymocytes[47]. Overall, these results suggest that HIVEP3 activity might be uniquely required by T cells that, in addition to strong TCR signaling during selection, also require homotypic interactions between signaling lymphocyte activation molecule family (SLAMF) receptors, such as SLAM and Ly108 whose signals are intracellularly propagated by the adapter molecule SAP[68]. Further experiments will be necessary to test for such possibility.

Starting from single-cell transcriptomic analyses of thymic iNKT cells, we uncovered a crucial and unique function for Hivep3 in regulating the common developmental path of $T_{inn}$ cells. Such findings further refine our fundamental understanding of $T_{inn}$ development, which given the strong conservation of $T_{inn}$ cells throughout evolution, might prove valuable when considering human diseases associated with $T_{inn}$ defects.

## Methods

**Mice**. The $Hivep3^{-/-}$ mice backcrossed to the C57BL/6 background have been described previously and were graciously provided by Dr. Laurie Glimcher. C57BL/6 and CD45.1 congenic C57BL/6 mice were purchased from Jackson Laboratories. Mice of both sexes were used between 8 to 10 weeks and were age-matched for each experiment. All mice were raised in a specific pathogen-free environment at the Office of Laboratory Animal Research at the University of Colorado Anschutz Medical campus, where they were maintained at 21 °C on a 12 h light-dark cycle (6 a.m. to 6 p.m.) and given free access to food and water. Studies, as well as the method of euthanasia were carried out in accordance with the US Department of Health and Human Services Guide for the Care and Use of Laboratory Animals and were performed under protocol (00065) approved by the Institutional Animal Care and Use Committee of the University of Colorado Anschutz Medical campus. To induce euthanasia, $CO_2$ was administered from a compressed gas tank for at least 1 min and mice were monitored for absence of heartbeat and respiration. As a secondary physical method of euthanasia, cervical dislocation was performed following $CO_2$ administration.

**Thymocyte isolation and flow cytometry**. Single cell suspensions were prepared from the thymus by manual disruption using a syringe plunger. PBS57-CD1d1 and 5-OP-RU-MR1 tetramers were obtained from the National Institutes of Health Tetramer Core Facility. The complete list of surface antibodies used is as follows: from BioLegend—CD3 (clone 17A2; Catalog #100204, Catalog #100218), CD4 (clone GK1.5; Catalog #100447), CD4 (clone RM4-5; Catalog #100557), CD5 (clone 53-7.3; Catalog #100624), CD19 (clone 6D5; Catalog #115509, Catalog #115521, Catalog #115528), CD25 (clone PC61; Catalog #102006), CD44 (clone IM7; Catalog #103044), CD45.1 (clone A20; Catalog #110714), CD45.2 (clone 104; Catalog #109827, Catalog #109828), CD69 (clone H1.2F3; Catalog #104530), CD81 (clone Eat-2; Catalog #104913), CD127 (clone SB/199; Catalog #121123), CD138 (clone 281-2; Catalog #142531), H-2K^b (clone AF6-88.5; Catalog #116513), ICOS (clone C398.4 A; Catalog #313530), ICOS (clone 7E.17G9; Catalog #117406, Catalog #117424), Izumo1r (also called FR-4; clone 12A5; Catalog #125012), Lineage cocktail (clones 145-2C11, RB6-8C5, RA3-6B2, Ter-119 and M1/70; Catalog #133303), Ly-6C (clone HK1.4; Catalog #128029, Catalog #128030), NK1.1 (clone PK136; Catalog #108713, Catalog #108730, Catalog #108736, Catalog #108753), PD-1 (clone 29 F.1A12; Catalog #135206, Catalog #135220), TCRβ (clone H57-597; Catalog #109220, Catalog #109224, Catalog #109230) and Vγ1 (clone 2.11; Catalog #141104, Catalog #141108, Catalog #141112); from BD Biosciences — CCR9 (clone CW-1.2; Catalog #565412), CD3ε (clone 145-2C11; Catalog #612771), CD4 (clone H129.19; Catalog #747275), CD8α (clone 53-6.7; Catalog #563786), CD24 (clone M1/69; Catalog #563545, Catalog #612953), CD122 (clone TM-β1; Catalog #562960), NKp46 (clone 29A1.4; Catalog #561169), TCRγδ (clone GL3; Catalog #563993) and Vδ6.3 (clone 8F4H7B7; Catalog #555321); from Thermo Fisher Scientific—CD122 (clone TM-β1; Catalog #50-245-876).

After surface antibody staining, the cells were fixed and permeabilized using the FoxP3 fixation/permeabilization kit (Thermo Fisher Scientific). Fixed and permeabilized cells were incubated with a combination of the following antibodies targeting intracellular proteins: from Abcam—Lef-1 (clone EPR2029Y; Catalog #ab137872); from BioLegend—Bcl-2 (clone BCL/10C4; Catalog #633512), BrdU (clone 3D4; Catalog #364110, Catalog #364114), GATA3 (clone 16E10A23; Catalog #653808), Helios (clone 22F6; Catalog #137222), IFNγ (clone XMG1.2; Catalog #505810, Catalog #505826, Catalog #505830), IL-4 (clone 11B11; Catalog #504104), PLZF (clone 9E12; Catalog #145808), Granzyme B (clone GB11; Catalog #515408) and T-bet (clone 4B10; Catalog #644816, Catalog #644824, Catalog #644835); from BD Biosciences—RORγt (clone Q31-378; Catalog #562684); from Miltenyi Biotec—Tox (clone REA473; Catalog #130-118-335); from Thermo Fisher Scientific—Egr2 (clone erongr2; Catalog #12-6691-82, Catalog #17-6691-82, Catalog #25-6691-82), FoxP3 (clone FJK-16s; Catalog #58-5773-82), Granzyme A (clone GzA-3G8.5; Catalog #17-5831-82), Granzyme B (clone NGZB; Catalog #48-8898-82), Ki-67 (clone SolA15; Catalog #25-5698-82, Catalog #48-5698-82),

Granzyme A (clone GzA-3G8.5, Catalog #17-5831-82), PLZF (clone Mags.21F7; Catalog #53-9320-82, Catalog #25-9322-82) and RORγt (clone B2D; Catalog #17-6981-82).

The stained cells were then analyzed on a BD LSRFortessa (BD Biosciences)) using the BD FacsDiva software (v8.0) or Cytek Aurora (Cytek) and data were processed with FlowJo software vX (TreeStar).

**Enrichment of CD1d reactive thymocytes.** Thymocytes were enriched for PBS57-CD1d reactive cells by incubating thymocyte cell suspensions with PE or APC conjugated PBS57-CD1d tetramers for 45 min at 4 °C, then incubated with anti-PE or anti-APC magnetic microbeads (Miltenyi Biotec) for 15 min at 4 °C, followed by separation by using an autoMACS Pro Separator (Miltenyi Biotec) according to manufacturer's instructions. Subsequently, cells in the positive fraction were first stained for surface markers followed by staining for intracellular markers before being subjected to flow cytometric acquisition.

**Lung single cell suspensions.** To prepare single-cell suspensions lungs were finely chopped with scissors in a 48 well plate and treated with 3 μg ml$^{-1}$ collagenase III (Worthington, Lakewood, NJ), 5 μg ml$^{-1}$ DNAse, and complete media (RPMI containing 10% fetal calf serum and enriched supplements) for 60 min at 37 °C with gentle pipetting at 20 min intervals. Cells were then filtered through a 70 μm cell strainer and washed with complete media. Approximately $2 \times 10^6$ cells were filtered (40 μm) and used for flow cytometric analysis.

**Liver single cell suspensions.** Hepatic leukocytes were isolated by cutting individual livers into small pieces and gently pressed through a 70 μm filter placed on top of a 50 ml falcon tube and resuspended in FACS buffer (PBS, 0.5% BSA, 0.5 mM EDTA, 1% Azide). The cells were washed twice in ice-cold FACS buffer and spun through 33.8% Percoll (Amersham Pharmacia Biotech) for 12 min at 2000 rpm, room temperature. Red blood cells were removed by resuspending the pellet in red blood cell lysis buffer for 5 min at room temperature and washed with complete media. Cells were resuspended in FACS buffer and filtered through a 40 μm cell strainer.

**Bone marrow chimeras.** Bone marrow cells from congenic C57BL/6 mice (CD45.1) and *Hivep3*$^{-/-}$ (CD45.2) mice were harvested and depleted of CD90-expressing cells using magnetic beads. Bone marrow cells from CD45.1 mice were mixed at 1:1 ratio with CD45.2 *Hivep3*$^{-/-}$ mice. $5 \times 10^6$ cells were injected intravenously into lethally-irradiated recipient mice (1000 rads). Eight to ten weeks after reconstitution, chimeric mice were euthanized and thymocytes were analyzed by flow cytometry.

**BrdU incorporation assay.** Mice received an intraperitoneal injection of BrdU (BrdUrd; 2 mg in 200 μL PBS; BD Biosciences) or PBS only at 0, 24, and 48 h before being euthanized for analysis at 72 h. To detect the incorporation of BrdU, the BrdU Flow Kit (BD Biosciences, Catalog #552598) was used according to the manufacturer's protocol.

**In vivo stimulation of iNKT cells.** Mice received 2 μg of αGC (Alexis Biochemicals) or vehicle by intravenous injection. Ninety minutes after injection, organs were collected and processed. Cells were first surface stained, fixed and subsequently stained intracellularly for IL-4 and IFNγ.

**scRNA-seq data processing.** The quality of sequencing reads was evaluated using FastQC and MultiQC. Cell Ranger v3.1.0 was used to align the sequencing reads (FASTQ) to the mm10 mouse transcriptome and quantify the expression of transcripts in each cell. This pipeline produced a gene expression matrix for each sample, which records the number of UMIs for each gene associated with each cell barcode. Unless otherwise stated, all downstream analyses were implemented using R v3.6.1 and the package Seurat v3.1.4 (ref. [69]). Low-quality cells were filtered using the cutoffs nFeature_RNA > = 500 & nFeature_RNA < 4100 & nCount_RNA > = 1000 & percent.mt < 6. The NormalizeData function was performed using default parameters to remove the differences in sequencing depth across cells. Dimension reduction was performed at three stages of the analysis: the selection of variable genes, PCA, and uniform manifold approximation and projection (UMAP)[70]. The FindVariableGenes function was applied to select highly variable genes covering most biological information contained in the whole transcriptome. To remove batch-effect from the PCA subspaces based on the correct cell alignment, we used fastMNN[16] to detect mutual nearest neighbors (MNN) of cells in different batches, and then used the MNN to correct the values in each PCA subspace. The variable genes were used for MNN as input to perform the RunUMAP function to obtain bidimensional coordinates for each cell. We determined the k-nearest neighbors of each cell using the FindNeighbors function and used this knn graph to construct the SNN graph by calculating the neighborhood overlap (Jaccard index) between every cell and its k.param nearest neighbors. Finally, we used the FindClusters function (resolution 0.8) to cluster cells using the Louvain algorithm based on the same PCs as RunUMAP function.

**Identification of differentially expressed genes.** We identified cluster-enriched genes by using the FindAllMarkers function in Seurat and the Wilcoxon-Rank sum test. This function identified differentially expressed genes for each cluster by comparing the gene expression for cells belonging to a cluster versus cells belonging to all other clusters. Only those genes that passed an adjusted *p* value (Benjamini–Hochberg) cutoff of 0.05 were included in the downstream analyses. Gene ontology (GO) analysis was performed by using the R package clusterProfiler (v3.0.4)[71].

**Developmental trajectory inference.** Two independent trajectory packages (Slingshot and Monocle v3) were used to order the cells in pseudotime. To run Slingshot, we first converted the Seurat object into a SingleCellExperiment object. We then implemented the Slingshot algorithm in a semi-supervised manner by providing this object as input to the slingshot function and specifying the start cluster as cluster 0 (start.clus = '0'). This function then determines the global lineage structure using cluster-based minimum spanning tree and subsequently fitting simultaneous principal pseudotime curves to describe each determined lineage. Lastly, we plotted the lineage curves on the UMAP plot generated by the Seurat package. For the Monocle 3 pipeline, we followed the recommended workflow to create a cell_data_set object with our cells. Since the UMAP coordinates for the cells provided by Monocle 3 were not equivalent to the coordinates provided by Seurat, we replaced the UMAP coordinates for each cell in the cell_data_set object with those from the Seurat object. We then ordered the cells using the order_cells function and choosing cluster 0 cells as the root node. The trajectories were then plotted on the UMAP using the plot_cells function and setting the color_cells_by argument to 'pseudotime'.

**ATAC-Seq.** ATAC-seq was performed according to the Omni-ATAC protocol previously described by Corces et al.[72]. 5,000 to 20,000 cells from each iNKT subset (iNKT1 cells (CD122$^+$, ICOS$^-$), iNKT2 cells (CD122$^-$, ICOS$^+$, Izumo1r$^+$, CD138$^-$) or iNKT17 cells (CD122$^-$, ICOS$^+$, Izumo1r$^-$, CD138$^+$)) from the pooled thymi of C57BL/6 or *Hivep3*$^{-/-}$ mice were sorted individually and then pooled for a total of 50,000-100,000 cells per transposition. Cells were then placed in 50 μL of cold lysis buffer (10 mM TrisHCl, pH 7.4, 10 mM NaCl, 3 mM MgCl2, 0.1% (v/v) Molecular biology-grade NP-40, 0.1% (v/v) Tween-20, 0.01% (v/v) Digitonin). Permeabilized cells were pelleted for 10 min at 500xg and resuspended in 50 μL of transposition mix (25 μL 2x TD buffer (Illumina), 16.5 μL 1X PBS, 0.5 μL 10% (v/v) Tween-20, 0.5 μL 1% (v/v) Digitonin, 2.5 μL Tn5 transposase (Illumina), 5 μL nuclease-free water). The volume of lysis buffer/transposition mix was scaled up to 100 μL for 100,000 cells. The transposase reaction was conducted for 30 min at 37 °C with shaking at 1,000 rpm. Cells were then subjected to a second sort to separate the different iNKT cell subsets using the same markers used in the first sort. While the fluorescence corresponding to the CD122 and Izumo1r markers was quenched by the transposition reaction, other markers were unaffected, allowing for the sorting of each of the iNKT cell subsets on the basis of ICOS and CD138 expression (iNKT1 (ICOS$^-$ CD138$^-$), iNKT2 (ICOS$^+$, CD138$^-$) and iNKT17 (ICOS$^+$ CD138$^+$)). Chromatin accessibility surrounding subset-defining genes (i.e. *Tbx21*, *Rorc* and *Slamf6*) is in accordance with the expected pattern of each iNKT subset (Supplementary Fig. 6). DNA was purified using the DNA Clean and Concentrator-5 kit (Zymo Research) and barcoding was performed using Illumina compatible index primers designed according to Buenrostro et al.[73] (Integrated DNA Technologies). PCR was conducted for 11–12 cycles. Library purification and size selection was carried out using 1.8X AMPureXP beads (Beckman Coulter). Libraries were quantified and size distribution was assessed using the High Sensitivity D1000 ScreenTape System (Agilent). Paired-end sequencing was performed on a NovaSeq (Illumina) with 150 cycles for each read. Raw data from the sequencer was demultiplexed and FASTQ files were generated using bcl2fastq conversion software (Illumina).

**ATAC-Seq analysis.** Sequencing reads in FASTQ format were trimmed to 50 bp using Trimmomatic and mapped to mouse genome (mm10) using the ENCODE ATAC-seq pipeline, with default parameters, except mapped reads were sub-sampled to a maximum of 50 million reads for peak calling. NarrowPeak files from individual replicates of all conditions were merged into a global set of peaks, excluding those on the Y chromosome or overlapped the ENCODE blacklist, and condensed to non-overlapping regions with a uniform size of 500 bp using chromVAR[74]. The number of transposase insertions within each region was computed for each replicate using chromVAR and these raw ATAC-seq counts per peak for all replicates were normalized using voom. Pairwise contrasts were performed with limma and differentially accessible regions were filtered based on an FDR adjusted p-value of less than 0.1 and an estimated fold-change of at least 3. We computed the ATAC-seq density (number of transposase insertion sites per kilobase) and accessible regions were defined as those with a mean of 5 normalized insertions per kilobase. We associated transcription factors binding motifs from the HOMER database by determining the enrichment of motifs in groups of peaks with HOMER and comparing the variability in ATAC-seq signal with chromVAR. For visualization, genomic coverage for individual replicates were computed on 10 bp windows with MEDIPS using full fragments captured by ATAC-seq and used to generate average coverage with the Java Genomics Toolkit for each group.

**Statistical analysis**. Data are presented as mean ± SD (with dots representing individual values). Statistical differences between individual groups were calculated using the two-tailed Mann–Whitney $U$-test using Prism 8.0 (GraphPad Software Inc.) with $*p < 0.05$, $**p < 0.01$, $***p < 0.001$ and $****p < 0.0001$.

**Reporting summary**. Further information on research design is available in the Nature Research Reporting Summary linked to this article.

## Data availability

Data that support the findings of this study have been deposited in NCBI GEO (http://www.ncbi.nlm.nih.gov/geo/) with the accession codes GSE152786 and GSE160518. All the other data are included within the article, or supplemental information or available from the authors upon reasonable requests. Source data are provided with this paper.

## Code availability

All the custom code used for processing and analyzing the data in the study were compiled into a single publicly available GitHub repository https://github.com/krovi137/NKT-Single-Cell-Analysis/tree/main.

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

## Acknowledgements

We thank members of our laboratories for thoughtful discussions and critical comments on the manuscript; Jennifer Matsuda and Leslie Berg for critical comments and support; the Flow Core and the University of Colorado flow cytometry shared resource facility for assistance with cell sorting; the Genomics and Microarray core at the University of Colorado Anschutz Medical campus for scRNA-Seq and ATAC-seq and the National Institutes of Health core facility for CD1d and MR1-tetramers. This work was supported by National Institutes of Health Grants AI135339 and AI130198 (to L.G.); The Cancer Center Support Grant P30CA046934.

## Author contributions

L.G. and S.H.K. wrote the manuscript and designed figures. L.G., S.H.K., J.Z. and J.S.B. designed experiments. S.H.K., J.Z., M.J., M.-F. and L.L. conducted experiments and acquired data. S.H.K., J.Z., M.J.M-F., L.L., T.B., J.S.B. and L.G. analyzed and interpreted data. L.G., S.H.K., and J.S.B. edited the manuscript. All authors approved the final manuscript.

## Competing interests

The authors declare no competing interests.
