## [Peer Review File · Nature Communications]

REVIEWER COMMENTS

Reviewer #1 (NKT, MAIT, atypical T cells.) (Remarks to the Author):

This paper represents an extensive single cell analysis of the transcriptome and chromatin accessibility of mature and immature NKT thymocytes. The authors then compare their data to those previously published on MAIT cells to confirm the similarities in the development of these two subsets at the single cell level. This is further confirmed by the similar effect of HIVEP inactivation on the number of NKT, MAIT cells and PLZF expressing gd T cells.

The main new findings are the following:

- A better description of the initial phase of NKT development and the absence of a defined precursor that had previously been suggested,
- The NKT2 subsets may not be a terminally differentiated effector subsets but rather an intermediate in the differentiation of both NKT1 and NKT17 subsets
- The similarities between the NKT and MAIT development pathway
- Demonstration that the HIVEP transcription factor is exclusively necessary of the accumulation of the PLZF expressing T cell subsets, NKT, MAIT and gd T cells.

The findings are original and add to the field. The study is well conceived and the quality of the data and analysis is excellent. The manuscript is easy to read and both the main figures and supplementary ones are highly legible and interesting. Although long, the discussion adds to the results by providing the consequences of the findings and the possible mechanisms involved. The discussion suggests follow-up experiments to decipher the different hypothesis. However, these experiments are beyond the scope of this manuscript which represents already an extensive and nice piece of work.

A few minor questions:

- In fig.1 and 2, CD69 is expressed not only at the initial phase of the differentiation but also in NKT1 mature subsets. This would suggest encounter with antigen. This is surprising because the mature NKT1 cells have probably moved to the medulla away from the CD1d expressing DP thymocytes. This could be discussed in the context of Wang et al 2019 already cited.
- Strikingly, the NKT1/2/17 subset distribution is not modified in HIVEP KO mice suggesting that HIVEP is operating after subset commitment which seems to happen at the initial phase of differentiation as suggested by this group (Tuttle et al). What is the TCRb Vb2/7/8 distribution in the NKT1/2/17 subsets.

Reviewer #2 (Transcriptional regulation of immune cell development.) (Remarks to the Author):

Single cell analysis of thymic iNKT cells unmasks the common developmental program of innate T cells and a shared requirement for the transcription factor HIVEP3.

In this manuscript the transcriptional landscape of iNKT cell maturation and fate decisions under steady-state conditions was investigated. The data reveals the transcriptional signature of the previously identified stage 0, iNKT1, iNKT2 and iNKT17 cells, and further expose heterogeneity amongst iNKT1 cells. The data also uncover a transcriptional signature shared between iNKT and MAIT cells, revealing a common transcriptional circuitry that underpins the development of these two lineages.

Notably, the investigators identify a requirement for a zinc finger transcription factor named Hivep3 in modulating iNKT cell development. In sum, these observations identify Hivep3 as a novel regulator of

iNKT cell development.

This is an important and well-done study. The data are compelling and of interest to the immunology readership. I recommend publication in Nature Communications.

Comments:

1. Could the authors please compare the distribution of transcripts obtained from scRNA-seq analysis to that derived from bulk RNA-seq?
2. The Discussion suffers from verbosity. The manuscript would be greatly improved if it would simply highlight important findings and briefly discuss this in the context of previous work.
3. Not much of an attempt is made to link Hivep3 to other transcription factors that control iNKT cell development. A diagram of putative links involving Hivep3 and other factors would be helpful.

Reviewer #3 (Unconventional T, NKT, gd T.) (Remarks to the Author):

Krovi et al. undertake to examine the development of innate-like T cells in the thymus using an unbiased single cell transcriptomic approach. Using invariant iNKT cells as a model, the authors profiled >10,000 cells, identifying a number of unique clusters based on transcriptional landscape and, subsequently, developmental pathway. In a comprehensive analysis, a developmental pathway from a previously identified stage 0 phenotype cells, through to terminally differentiated iNKT1, iNKT2 and iNKT17 clusters is outlined based on transcription of genes known to be important for these sub-populations. In the course of this analysis, Hivep3 is identified as a transcriptional regulator expressed predominantly in stage 0 cells. Hivep3^{-/-} mice were examined and found to contain very few developing iNKT cells, due to a cell intrinsic block in development, largely at the stage 0 to stage 1 developmental step. HIVEP3 alters transcription in developing iNKT cells via a number of mechanisms, including chromatin remodelling, its action as a transcriptional adaptor, as well as possible secondary transcriptional effects. The authors then examine the role of HIVEP3 in other PLZF-dependent innate-like populations, including MAIT cells, $\gamma\delta$ T cells and ILCs, and PLZF-independent 'agonist-selected' cells, such as Tregs and IELs. The absence of HIVEP appears to have a specific effect on PLZF-dependent innate-like T cells, and as such suggests a synergy between HIVEP3 and PLZF in innate-like T cell development.

Overall Krovi et al. provide a nuanced, and detailed investigation into the development of iNKT cells, with compelling evidence for a role of HIVEP3 in iNKT cells and, to a lesser extent, other innate-like T cells. There are a number of points that could help the reader follow what is generally a well laid out, but data-heavy manuscript.

General comments;

1. NKT cell development has been described as a three-stage process that the authors use in some sections, but complicating this is the existence of iNKT1, iNKT2 and iNKT17 cells that the authors use in other sections. Cells that fall into the iNKT2 bin are especially difficult to place in the developmental sequence because many of these are precursors or other populations rather than mature committed cells (Benlagha et al Science 2002; Pellicci et al JEM 2002; Gadue et al JI 2002). The authors have carefully analysed iNKT2 cells and indeed provide data that many of these appear to be in a transitional state bridging towards iNKT1 and iNKT17 cells. However, discussion of this issue could be improved as it is hard to follow the thread of the argument in places. In particular, when discussing the developmental pathway suggested by unsupervised clustering, and later pseudo-time inference,

referring to clusters 1, 2 and 3 as iNKT2 cells based on Gata3 and Il4 expression is confusing; especially given their resemblance to iNKTp cells both by transcriptional signature and entry in to cell cycle/proliferative expansion. Based on a reasonable level of evidence that the authors provide, it is likely that at least clusters 1-3 represent a developmental step after stage 0 that does not solely lead to iNKT2 cells but is an intermediate in iNKT cell development generally and can be labelled as such (i.e. iNKTp cells), which reflect the previously defined stage 1 and 2 iNKT cells. The classification of iNKT2 cells is a problematic area in iNKT cell development. Here, the authors have an opportunity to provide some clarity toward this problem. It would be more valuable to attempt to distinguish precursor populations from mature "bona fide" iNKT2 cells.

2. Following on from point 1, given the discussion focused around the developmental pathway is subtle and largely explores potential pitfalls and inconsistencies with the literature, I believe it would be of great use to the field to spell-out a more definitive pathway based on the data in the manuscript and further correlate this pathway with current the understanding of stage 0, 1, 2 and 3. I.e. cluster 0 (stage 0) \diamond cluster 1, 2, 3 (iNKTp or stage 1 and 2) \diamond cluster 4 (iNKT2) or cluster 5 (iNKT17) or cluster 6 (iNKT1 intermediate?). Cluster 6 \diamond cluster 8 \diamond cluster 9, 10. Cluster 7 different population of iNKT1 or another intermediate?

3. To fully appreciate the author's thesis, the reader is compelled to understand the transcriptional profiles of the clusters and how they relate to one-another. However, as each unsupervised clustering is repeated to include additional data the clusters reset, as do the UMAP graphs, making the reader re-learn what each cluster refers to for each analysis. Whilst the nature of performing unsupervised clustering results in this reset, to aid the reader can a seed value be used to provide uniformity between analyses? Or if seeding is not possible with the particular programme, can all the data be included in a single analysis, and only the pertinent data displayed for each section? This will help the reader follow the similarities and differences between each analysis with greater ease. E.g. there are 11 clusters with just the B6 iNKT cells, but only 10 with the B6 and Hivep3^{-/-} iNKT cells, what does that missing cluster relate to?

4. It is difficult to work through data when UMAP graphs are solely provided when identifying specific clusters certain genes appear in. It would be very helpful if the authors could provide summary graphs, or gates with labels on the UMAP graph, or at least one labelled UMAP graph beside each figure so the take home message is clear and does not require referring back to previous figure to decode the information. Especially Fig. 2d, 3b, and 6.

5. Figure 7 and related results section:

The authors have homed into a link between HIVEP3 and strong TCR signalling required in the early stage 0 development of innate-T cells (and subsequently PLZF expression), with TCR and CD3 related genes implied as DEGS Fig.5. Fig. 7 draws upon their previous expertise (Tuttle et al 2018) to examine how HIVEP3 regulates chromatin accessibility. This begs the question of why wasn't ATAC seq performed on precursors upstream of iNKT1/2/17? (C57BL/6 and Hivep3^{-/-}-iNKTp's??)

Minor comments;

1. It would have been very helpful if a figure number was included on figures so reviewers can work out quickly which figure is which. Moreover, Sup. Fig. 4 is missing from the manuscript.

2. "While a large body of knowledge is available on iNKT cell characteristics and development in the thymus and in the periphery (Krovi and Gapin, 2018), the developmental steps underlying iNKT cell differentiation, and by extension MAIT cells, remain, nevertheless, incomplete."

3. Please elaborate on why Hivep3 was a good candidate to follow up.

4. Figure 4;

- Sometimes the dots are very hard to see. Maybe use large dots for part a, especially thymus where number of dots is very low.
- Part a, CD3⁻ cells have been gated out of LN and lung but not thymus, spleen and liver, why? And do the %s refer to frequency of T cells or all lymphocytes? The bi-ex axes are different between LN and lung, and thymus, spleen and liver, please use same width-basis scaling. Also, spleen has an uptick of CD3^{lo}/tetramer^{lo} cells, there might be a compensation issue there.
- There is a partial rescued of iNKT cell numbers in LN and lung. Are there peripheral mechanisms that can restore iNKT cell numbers?
- There is a lot of dead space on FACS plots from part d and f. It would be better to choose more appropriate width-basis scaling.
- Part e, please state what % are out of on summary graphs.

5. NKT cells, and CD44 cells more generally, are resistant to radio-ablation (Yao et al., JI, 2011). Given the transferred wild-type cells are indistinguishable from residual radio-resistant cells, this complicates interpretation. The more appropriate control is to transfer 50:50 CD45.1 and CD45.2 wild-type cells in to CD45.1 irradiated host, and examine the capacity of the CD45.2 wild-type cells to reconstitute the iNKT cell niche, not the CD45.1 wild-type cells from the Hivep3^{-/-} transfer. This excludes the possibility that the Hivep3 cells are absent, not because the defect is cell intrinsic, but because they cannot outcompete a niche occupied by the radio-resistant CD45.1 host cells.

6. "Nevertheless, we noticed that the remaining iNKT cells in Hivep3^{-/-} mice had overall higher TCR expression levels (Fig 4a, d) and that PLZF protein levels in each of the iNKT subsets were reduced (Fig 4f), ..." please quantify MFI, not refer to a representative dot plot with unbalanced axes.

7. "Altogether, these results show that in the absence of Hivep3, iNKT development is blocked at stage 0, with a few cells able to further progress in an aberrant fashion, bypassing the proliferative burst and instead acquiring an iNKT1 transcriptional signature earlier during development." Apart from gross changes to proportions of clusters, or overall expression of genes, are there specific transcriptional changes within individual clusters that might give a clue to the altered developmental pathway? E.g. do the iNKT1 cells that do develop have an alerted transcriptional signature, or do the cells that do develop have altered TCR usage?

8. Why was a relatively high padj of < 0.1 used?

9. The samples often do not satisfy the assumptions of student t test or ANOVA, being normal distribution and equal variance. Eg. Fig 4bceg, Fig 8f. Most figure legends lack information about reproducibility; Number of tests, mice, experiments. This should be listed for all figures.

10. Figure 7;

- Part d, there is significant variation between replicates, notably in the NF- κ B TF family within iNKT1 (Hivep3^{-/-}) and iNKT17 cells (B6), but also bZIP iNKT1 (Hivep3^{-/-}) iNKT17 (B6), and ETS iNKT2 (B6)

11. Figure 8;

- Parts e and g lungs, it would be helpful to choose a more appropriate bi-ex width-basis scaling
- Part h, MFI would be a better measure of PLZF expression
- Part g, representative plot for Vg1Vd6.3 cells in spleen shows the frequency the same between B6 and Hivep3^{-/-}. In the text Vg1Vd6.3 cells are described as lower in all organs in Hivep3^{-/-} mice.

12. Capitalisation of 'i' in iNKT "Early CD24+ MAIT cells progenitors (in cluster 0) share expression of many genes with Stage 0 INKT iNKT cells, including Itm2a, Cd24a and Hivep3 ((Fig 8c, d and Sup. Table III)."

13. Gene name not italicised "By binding to TRAF2 (Oukka et al., 2002), HIVEP3 can inhibit both NF-kB and c-Jun NH2-terminal kinase (JNK)-mediated responses, while by physically interacting with c-Jun it serves as a co-activator of AP-1-dependent IL-2 IL2 gene..."

14. "Upon thymic selection, HDAC7 is exported out of the nucleus in a TCR-dependent manner following TCR signalling..." - TCR does not remove HDAC7 from the nucleus.

15. When referring to impaired MAIT cell development in SAPKO mice it is appropriate to also cite Koay et al 2019 Science Immunology which also provided insight into SLAM family members that are DEGs during MAIT development.

Reviewer #1 (Remarks to the Author):

This paper represents an extensive single cell analysis of the transcriptome and chromatin accessibility of mature and immature NKT thymocytes. The authors then compare their data to those previously published on MAIT cells to confirm the similarities in the development of these two subsets at the single cell level. This is further confirmed by the similar effect of HIVEP inactivation on the number of NKT, MAIT cells and PLZF expressing gd T cells.

The main new findings are the following:

- A better description of the initial phase of NKT development and the absence of a defined precursor that had previously been suggested,
- The NKT2 subsets may not be a terminally differentiated effector subsets but rather an intermediate in the differentiation of both NKT1 and NKT17 subsets
- The similarities between the NKT and MAIT development pathway
- Demonstration that the HIVEP transcription factor is exclusively necessary of the accumulation of the PLZF expressing T cell subsets, NKT, MAIT and gd T cells.

The findings are original and add to the field. The study is well conceived and the quality of the data and analysis is excellent. The manuscript is easy to read and both the main figures and supplementary ones are highly legible and interesting. Although long, the discussion adds to the results by providing the consequences of the findings and the possible mechanisms involved. The discussion suggests follow-up experiments to decipher the different hypothesis. However, these experiments are beyond the scope of this manuscript which represents already an extensive and nice piece of work.

We would like to thank the reviewer for his/her time and for considering our manuscript as well conceived with high quality data and analysis.

A few minor questions:

- In fig.1 and 2, CD69 is expressed not only at the initial phase of the differentiation but also in NKT1 mature subsets. This would suggest encounter with antigen. This is surprising because the mature NKT1 cells have probably moved to the medulla away from the CD1d expressing DP thymocytes. This could be discussed in the context of Wang et al 2019 already cited.

As pointed out by the reviewer, CD69 transcripts (Fig 1d) and protein (Fig 1f) are detected in thymic stage 0 as well as thymic iNKT1 cells. Although this likely suggests that iNKT1 are receiving signaling, it is unclear whether for iNKT1 this signaling is mediated by the TCR through the recognition of CD1d + antigen. Apart from CD69, we do not find evidence for further TCR signaling of iNKT1 cells at the transcriptional level, which should be apparent through the induction of multiple transcription factors known to be downstream of TCR signaling, such as LEF1, Nr4a1 etc. Instead we favor the possibility that CD69 upregulation by iNKT1 might be induced through cytokine signaling. For example, IL-15 has been previously shown to upregulate CD69 in cells sensitive to this cytokine (Lin et al. 2000) and iNKT1, which express CD122, certainly fall into this category. Similarly, CD69 upregulation on T lymphocytes can be induced upon type I interferon signaling (Alsharifi, 2005).

- Strikingly, the NKT1/2/17 subset distribution is not modified in HIVEP KO mice suggesting that HIVEP is operating after subset commitment which seems to happen at the initial phase of differentiation as suggested by this group (Tuttle et al). What is the TCRb Vb2/7/8 distribution in the NKT1/2/17 subsets.

We analyzed the V β distribution amongst iNKT subsets from 3 B6 and 3 Hivp3 KO mice and did not detect notable changes between wildtype and Hivp3-deficient mice (Tukey's multiple comparisons test, only B6 vs Hivp3

comparison are shown for clarity). These results suggest that Hivp3 is affecting the commitment of the cells to the iNKT lineage but not their subset differentiation.

Reviewer #2 (Remarks to the Author):

Single cell analysis of thymic iNKT cells unmask the common developmental program of innate T cells and a shared requirement for the transcription factor HIVEP3.

In this manuscript the transcriptional landscape of iNKT cell maturation and fate decisions under steady-state conditions was investigated. The data reveals the transcriptional signature of the previously identified stage 0, iNKT1, iNKT2 and iNKT17 cells, and further expose heterogeneity amongst iNKT1 cells. The data also uncover a transcriptional signature shared between iNKT and MAIT cells, revealing a common transcriptional circuitry that underpins the development of these two lineages. Notably, the investigators identify a requirement for a zinc finger transcription factor named HIVEP3 in modulating iNKT cell development. In sum, these observations identify HIVEP3 as a novel regulator of iNKT cell development.

This is an important and well-done study. The data are compelling and of interest to the immunology readership. I recommend publication in Nature Communications.

We thank the reviewer for their kind comments regarding our manuscript.

Comments:

1. Could the authors please compare the distribution of transcripts obtained from scRNA-seq analysis to that derived from bulk RNA-seq?

We extracted the gene signature of each iNKT cell subset obtained from bulk RNA-seq as described in Engel et al. and displayed individual single-cell heatmap for each of the genes within these signatures where the columns indicate row-wise Z-scores from 100 single cells in each of the clusters identified. While there is substantial overlap between the subset specific signatures obtained by bulk RNA-seq and the cell identity that we ascribed to each cluster, the single cell data further suggest that the subset populations themselves are heterogeneous and studying them in bulk could be misleading.

2. The Discussion suffers from verbosity. The manuscript would be greatly improved if it would simply highlight important findings and briefly discuss this in the context of previous work.

We have now re-written part of the discussion highlighting the major findings of our study.

3. Not much of an attempt is made to link Hivep3 to other transcription factors that control iNKT cell development. A diagram of putative links involving Hivep3 and other factors would be helpful.

We agree with the reviewer and it represents an interesting point. However, we feel that any attempt to link Hivep3 to any other transcription factor activity would be highly speculative at this point. Many transcription factors have been shown to affect the development of iNKT cells over the years and it remains unclear how to meaningfully integrate all of these data. What TF regulates what genes and/or other TF and which ones interact with each other to regulate gene expression are important questions that remain mostly unanswered. The paucity of cells has certainly limited the number of biochemical studies looking at protein interactions directly within iNKT cells. This remains a bottleneck in iNKT cell studies.

Reviewer #3 (Remarks to the Author):

Krovi et al. undertake to examine the development of innate-like T cells in the thymus using an un-biased single cell transcriptomic approach. Using invariant iNKT cells as a model, the authors profiled >10,000 cells, identifying a number of unique clusters based on transcriptional landscape and, subsequently, developmental pathway. In a comprehensive analysis, a developmental pathway from a previously identified stage 0 phenotype cells, through to terminally differentiated iNKT1, iNKT2 and iNKT17 clusters is outlined based on transcription of genes known to be important for these sub-populations. In the course of this analysis, Hivep3 is identified as a transcriptional regulator expressed predominantly in stage 0 cells. Hivep3^{-/-} mice were examined and found to contain very few developing iNKT cells, due to a cell intrinsic block in development, largely at the stage 0 to stage 1 developmental step. HIVEP3 alters transcription in developing iNKT cells via a number of mechanisms, including chromatin remodelling, its action as a transcriptional adaptor, as well as possible secondary transcriptional effects. The authors then examine the role of HIVEP3 in other PLZF-dependent innate-like populations, including MAIT cells, $\gamma\delta$ T cells and ILCs, and PLZF-independent 'agonist-selected' cells, such as Tregs and IELs. The absence of HIVEP appears to have a specific effect on PLZF-dependent innate-like T cells, and as such suggests a synergy between HIVEP3 and PLZF in innate-like T cell development.

Overall Krovi et al. provide a nuanced, and detailed investigation into the development of iNKT cells, with compelling evidence for a role of HIVEP3 in iNKT cells and, to a lesser extent, other innate-like T cells. There are a number of points that could help the reader follow what is generally a well laid out, but data-heavy manuscript.

We thank the reviewer for his/her meticulous review of our manuscript.

General comments;

1. NKT cell development has been described as a three-stage process that the authors use in some sections, but complicating this is the existence of iNKT1, iNKT2 and iNKT17 cells that the authors use in other sections. Cells that fall into the iNKT2 bin are especially difficult to place in the developmental sequence because many of these are precursors or other populations rather than mature committed cells (Benlagha et al Science 2002; Pellicci et al JEM 2002; Gadue et al JI 2002). The authors have carefully analysed iNKT2 cells and indeed provide data that many of these appear to be in a transitional state bridging towards iNKT1 and iNKT17 cells. However, discussion of this issue could be improved as it is hard to follow the thread of the argument in places. In particular, when discussing the developmental pathway suggested by unsupervised clustering, and later pseudo-time inference, referring to clusters 1, 2 and 3 as iNKT2 cells based on Gata3 and Il4 expression is confusing; especially given their resemblance to iNKTp cells both by transcriptional signature and entry in to cell cycle/proliferative expansion. Based on a reasonable level of evidence that the authors provide, it is likely that at least clusters 1-3 represent a developmental step after stage 0 that does not solely lead to iNKT2 cells but is an intermediate in iNKT cell development generally and can be labelled as such (i.e. iNKTp cells), which reflect the previously defined stage 1 and 2 iNKT cells. The classification of iNKT2 cells is a problematic area in iNKT cell development. Here, the authors have an opportunity to provide some clarity toward this problem. It would be more valuable to attempt to distinguish precursor populations from mature "bona fide" iNKT2 cells.

We thank the reviewer for his/her summary of the presented data and we agree that the definition of intermediary developmental stages is complicated, particularly the difference between iNKTp and iNKT2 cells. We have attempted throughout most of the manuscript to display results based on the aforementioned three stages of development. However, because thymic iNKT cells can also be segregated on the basis of differential transcription factor expression, we also used these parameters in a few sections. While our manuscript was under consideration Paget's group reported single cell RNA-

Seq data from thymic iNKT cells (Baranek et al. 2020) and argued for a similar developmental pathway. Both sets of data are in agreement (see figure). Furthermore, to better define "bona fide" iNKT2 cells and to distinguish them from precursor populations, we have sorted IL-4-secreting iNKT2 cells ($CD122^{neg}$, $ICOS^{pos}$, $CD138^{neg}$) and non-IL4-secreting iNKT2 cells using the KN2 IL4-reporter mice and performed sc-RNA-seq on both samples. We then integrated the data with the B6 samples and determined how each sample would distribute on the UMAP. The data demonstrate that both samples contain multiple clusters based on differential gene expression. The relative proportions of cells in each cluster is different between both samples. IL-4 negative iNKT2 contains more cells that are on the developmental path to become iNKT1 or iNKT17 compared to IL-4 positive iNKT2 cells. Nevertheless, both samples share essentially the same clusters, with minimal differences in gene expression between the two samples. Thus, even using the accepted marker of bona-fide iNKT2 cells (ie. IL-4 production), it remains challenging to formally separate iNKT2 from precursors. These data are provided to the reviewer for illustration purpose only as we do not believe that they pertain to the current manuscript, which is already defined as "data heavy".

2. Following on from point 1, given the discussion focused around the developmental pathway is subtle and largely explores potential pitfalls and inconsistencies with the literature, I believe it would be of great use to the field to spell-out a more definitive pathway based on the data in the manuscript and further correlate this pathway with current the understanding of stage 0, 1, 2 and 3. I.e. cluster 0 (stage 0) \diamond cluster 1, 2, 3 (iNKTp or stage 1 and 2) \diamond cluster 4 (iNKT2) or cluster 5 (iNKT17) or cluster 6 (iNKT1 intermediate?). Cluster 6 \diamond cluster 8 \diamond cluster 9, 10. Cluster 7 different population of iNKT1 or another intermediate?

We have now edited the discussion of the manuscript.

3. To fully appreciate the author's thesis, the reader is compelled to understand the transcriptional profiles of the clusters and how they relate to one-another. However, as each unsupervised clustering is repeated to include additional data the clusters reset, as do the UMAP graphs, making the reader re-learn what each cluster refers to for each analysis. Whilst the nature of performing unsupervised clustering results in this reset, to aid the reader can a seed value be used to provide uniformity between analyses? Or if seeding is not possible with the particular programme, can all the data be included in a single analysis, and only the pertinent data displayed for each section? This will help the reader follow the similarities and differences between each analysis with greater ease. E.g. there are 11 clusters with just the B6 iNKT cells, but only 10 with the B6 and Hivep3^{-/-} iNKT cells, what does that missing cluster relate to?

We apologize to the reviewer and we agree with him/her that each new analyze generate a slightly different UMAP. However, the overall architecture of each UMAP remains largely unchanged. Unfortunately, at this point, our software does not allow us to "seed" the clustering so as to always generate the same UMAP across different datasets, only to regenerate the same projection when re-analyzing the same dataset.. We did attempt the second suggestion of the reviewer (ie. to integrate all the datasets together and subsequently subset only the pertinent data for each section), however this is problematic. For example, as we described in the manuscript, Hivep3^{-/-} iNKT have an enrichment in cells defined as cluster 1 with an early iNKT1 signature. Such cluster was defined and contain a very small number of cells in our B6 samples only because the data were integrated with the Hivep3 samples. Such cluster is not defined when analyzing the B6 samples by themselves. The missing cluster pointed out by the reviewer correspond to the cycling cells. When analyzing the B6 samples by themselves, we can clearly delineate cells in different phase of the cells cycle (S vs G2). In the integrated data with the Hivep3 sample, these two separate clusters are now considered as only one.

4. It is difficult to work through data when UMAP graphs are solely provided when identifying specific clusters certain

genes appear in. It would be very helpful if the authors could provide summary graphs, or gates with labels on the UMAP graph, or at least one labelled UMAP graph beside each figure so the take home message is clear and does not require referring back to previous figure to decode the information. Especially Fig. 2d, 3b, and 6.

We now provide a summary UMAP on the figures 2b and 3b with gates to clearly depict the iNKT cell subsets that we identified based on canonical markers. Unfortunately, adding such a summary UMAP to figure 6 would compromise a lot of space in the figure.

5. Figure 7 and related results section:

The authors have homed into a link between HIVEP3 and strong TCR signalling required in the early stage 0 development of innate-T cells (and subsequently PLZF expression), with TCR and CD3 related genes implied as DEGS Fig.5. Fig. 7 draws upon their previous expertise (Tuttle et al 2018) to examine how HIVEP3 regulates chromatin accessibility. This begs the question of why wasn't ATAC seq performed on precursors upstream of iNKT1/2/17? (C57BL/6 and *Hivep3*^{-/-}-iNKTp's??)

The reviewer raises an interesting point; however it remains technically challenging to perform ATAC-seq on such a small number of cells. In the future, we do intend to perform single-cell ATAC-seq on these samples so as to define chromatin accessibility at the single cell level in the course of iNKT cell development. In doing so, we should be able to provide information regarding chromatin accessibility within stage 0 iNKT cells. However, we believe that this is beyond the scope of the current manuscript.

Minor comments;

1. It would have been very helpful if a figure number was included on figures so reviewers can work out quickly which figure is which. Moreover, Sup. Fig. 4 is missing from the manuscript.

We apologize for this oversight. The figures are now numbered and Sup. Fig 4 is included.

2. "While a large body of knowledge is available on iNKT cell characteristics and development in the thymus and in the periphery (Krovi and Gapin, 2018), the developmental steps underlying iNKT cell differentiation, and by extension MAIT cells, remain, nevertheless, incomplete."

This sentence has been edited to, "While a large body of knowledge is available regarding iNKT cell characteristics and development in the thymus and in the periphery^{4,5,13-15}, the developmental steps underlying iNKT cell differentiation, and by extension MAIT cells, remain incomplete", on page 4 of the manuscript.

3. Please elaborate on why *Hivep3* was a good candidate to follow up.

While our data identified several transcription factors as highly expressed in stage 0 iNKT cells, a large number of them have already been reported to affect the development of the cells (*Lef1*, *Tox*, *Myb*, *Sox4*, *Id3*, *Egr1/2*), we therefore elected to focus on *Hivep3* for which little information currently exists regarding its role in the immune system.

4. Figure 4;

- Sometimes the dots are very hard to see. Maybe use large dots for part a, especially thymus where number of dots is very low.
- Part a, CD3⁻ cells have been gated out of LN and lung but not thymus, spleen and liver, why? And do the %s refer to frequency of T cells or all lymphocytes? The bi-ex axes are different between LN and lung, and thymus, spleen and liver, please use same width-basis scaling. Also, spleen has an uptick of CD3^{lo}/tetramer^{lo} cells, there might be a compensation issue there.
- There is a partial rescued of iNKT cell numbers in LN and lung. Are there peripheral mechanisms that can restore iNKT cell numbers?
- There is a lot of dead space on FACS plots from part d and f. It would be better to choose more appropriate width-basis scaling.
- Part e, please state what % are out of on summary graphs.

We have fixed the figure as pointed out by the reviewer. Most of our flow cytometry data were generated using spectral flow cytometers which adds an extra decade of fluorescence compared to "regular" flow cytometers. We have re-analyzed the data and removed the dead space. Some of the data (lungs and LN versus thymus, spleen

and liver) were generated on different days, with different animals and on different machines. It is expected that the scales of the flow plots will be slightly different. The “uptick” of tetramer staining in the spleen represents non-specific staining, which is common on splenocytes, and not a compensation problem. These “cells” were not gated nor considered as iNKT cells in our analysis.

5. NKT cells, and CD44 cells more generally, are resistant to radio-ablation (Yao et al., JI, 2011). Given the transferred wild-type cells are indistinguishable from residual radio-resistant cells, this complicates interpretation. The more appropriate control is to transfer 50:50 CD45.1 and CD45.2 wild-type cells in to CD45.1 irradiated host, and examine the capacity of the CD45.2 wild-type cells to reconstitute the iNKT cell niche, not the CD45.1 wild-type cells from the Hivep3^{-/-} transfer. This excludes the possibility that the Hivep3 cells are absent, not because the defect is cell intrinsic, but because they cannot outcompete a niche occupied by the radio-resistant CD45.1 host cells.

The reviewer raises an interesting point. Yao et al. examined the effects of total body irradiation 5 days post-irradiation. However, from past experiences in the reconstitution of the thymic iNKT cell compartment in bone marrow chimera mice with total irradiation recipients, reconstitution from the recipient bone marrow is extremely minimal at 8 weeks post-reconstitution (Tuttle et al.). In the current situation, wildtype bone marrow cells, being from the recipient or the wildtype donor bone marrow, were able to reconstitute the iNKT cell compartment. Yet, donor bone marrow cells from the Hivep3^{-/-} mice, did not.

6. “Nevertheless, we noticed that the remaining iNKT cells in Hivep3^{-/-} mice had overall higher TCR expression levels (Fig 4a, d) and that PLZF protein levels in each of the iNKT subsets were reduced (Fig 4f), ...” please quantify MFI, not refer to a representative dot plot with unbalanced axes.

We have now re-analyzed the data as per the reviewer’s recommendation and quantified the geometric MFI of PLZF and TCR expression per stages of development. The results are displayed in Sup. Figure 5 and referred to in the manuscript on page 9 and 10.

7. “Altogether, these results show that in the absence of Hivep3, iNKT development is blocked at stage 0, with a few cells able to further progress in an aberrant fashion, bypassing the proliferative burst and instead acquiring an iNKT1 transcriptional signature earlier during development.” Apart from gross changes to proportions of clusters, or overall expression of genes, are there specific transcriptional changes within individual clusters that might give a clue to the altered developmental pathway? E.g. do the iNKT1 cells that do develop have an alerted transcriptional signature, or do the cells that do develop have altered TCR usage?

The description of DEGs in each cluster between B6 and Hivep3^{-/-} iNKT cells is described in Sup. Table II. Unfortunately, we did not identify a specific pathway that would explain the overall change in gene expression between the B6 and Hivep3^{-/-} iNKT cells. As pointed out in the manuscript a large proportion of these DEGs are expressed at higher levels in Hivep3-deficient cells than B6 cells, hinting at the possibility for a role of Drosha in regulating miRNA processing and thereby post-transcriptional regulation of gene expression.

8. Why was a relatively high padj of < 0.1 used?

We used the FDR method for multiple testing correction with a padj set to 0.1, which is not uncommon in analyzing ATAC-seq data. It sets the number of “falsely” identified differentially accessible regions to 10% of the differentially expressed group. So, out of 1000 differentially accessible regions, 100 would be “false.” Lowering this number would certainly be lowering the number of false positive but would also run the risk to lose some potentially true positive. We combine the slightly high adjusted p value with a stricter fold-change cutoff (fold-change of at least 3), so all regions have to meet both criteria.

9. The samples often do not satisfy the assumptions of student t test or ANOVA, being normal distribution and equal variance. Eg. Fig 4bceg, Fig 8f. Most figure legends lack information about reproducibility; Number of tests, mice, experiments. This should be listed for all figures.

All group comparisons were re-analyzed using the Mann-Whitney *U* rank sum test in Prism, with **p* < 0.05, ***p* < 0.01, ****p* < 0.001 and *****p* < 0.0001. Information about reproducibility was added to each figure legend.

10. Figure 7;

- Part d, there is significant variation between replicates, notably in the NF- κ B TF family within iNKT1 (Hivep3^{-/-}) and iNKT17 cells (B6), but also bZIP iNKT1 (Hivep3^{-/-}) iNKT17 (B6), and ETS iNKT2 (B6)

We agree with the reviewer, but some variability is to be expected from such data. We used replicates for each sample which is commonly used in ATAC-seq experiments. A brand-new experiment with several more replicates would be required to further validate our findings since simply adding a new sample to the current data set would not be sufficient due to batch effects. This would significantly delay publication of our manuscript and we do not believe that it affects the conclusions of the experiment or the manuscript as it stands.

11. Figure 8;

- Parts e and g lungs, it would be helpful to choose a more appropriate bi-ex width-basis scaling
- Part h, MFI would be a better measure of PLZF expression
- Part g, representative plot for V γ 1/V δ 6.3 cells in spleen shows the frequency the same between B6 and Hivep3^{-/-}. In the text V γ 1/V δ 6.3 cells are described as lower in all organs in Hivep3^{-/-} mice.

See our response to comment 4 above, regarding the display of flow cytometry plots. We quantified the proportions of V γ 1/V δ 6.3 cells that are PLZF⁺ showing that in absence of Hivep3 expression, these cells are largely decreased in all the organ examined. We reanalyzed the data and added more mice to the analysis. The spleen samples demonstrate a consistent decrease in the proportion of V γ 1/V δ 6.3 amongst $\gamma\delta$ T cells and the proportion of V γ 1/V δ 6.3 that are PLZF⁺ is also decreased in Hivep3 KO mice.

12. Capitalisation of 'i' in iNKT "Early CD24⁺ MAIT cells progenitors (in cluster 0) share expression of many genes with Stage 0 INKT iNKT cells, including Itm2a, Cd24a and Hivep3 ((Fig 8c, d and Sup. Table III)."

This has been corrected.

13. Gene name not italicised "By binding to TRAF2 (Oukka et al., 2002), HIVEP3 can inhibit both NF- κ B and c-Jun NH2-terminal kinase (JNK)-mediated responses, while by physically interacting with c-Jun it serves as a co-activator of AP-1-dependent Il-2 Il2 gene..."

The names were not italicized in these cases because we were referring to the proteins.

14. "Upon thymic selection, HDAC7 is exported out of the nucleus in a TCR-dependent manner following TCR signalling..." – TCR does not remove HDAC7 from the nucleus.

The activity of HDAC7 is controlled by nuclear exclusion in response to phosphorylation of conserved serine residues in their N-terminal adapter domains (Verdin et al., 2003). In thymocytes, TCR stimulation results in HDAC7 phosphorylation and nuclear exclusion via Protein Kinase D (Parra et al., 2005). We have edited the sentence on page 15 of the discussion.

15. When referring to impaired MAIT cell development in SAPKO mice it is appropriate to also cite Koay et al 2019 Science Immunology which also provided insight into SLAM family members that are DEGs during MAIT development.

This part of the discussion has been edited out and therefore the citation is not required anymore.

REVIEWERS' COMMENTS

Reviewer #1 (Remarks to the Author):

The authors successfully answered the different comments. Well done.

Reviewer #2 (Remarks to the Author):

My comments have been addressed. The manuscript is now suitable for publication.

Reviewer #3 (Remarks to the Author):

The authors have provided a satisfying set of replies to my review and I have no further comments other than to congratulate them on their revised version which I think is acceptable for publication.